# Hypoxia inducible factor signaling in breast tumors controls spontaneous tumor dissemination in a site-specific manner

Vera M. Todd[1,2], Lawrence A. Vecchi III[2,3], Miranda E. Clements[4], Katherine P. Snow[2,5], Cayla D. Ontko[6], Lauren Himmel [7], Christopher Pinelli[7], Marjan Rafat [2,8] & Rachelle W. Johnson [1,2,3✉]

Hypoxia is a common feature in tumors and induces signaling that promotes tumor cell survival, invasion, and metastasis, but the impact of hypoxia inducible factor (HIF) signaling in the primary tumor on dissemination to bone in particular remains unclear. To better understand the contributions of hypoxia inducible factor 1 alpha (HIF1α), HIF2α, and general HIF pathway activation in metastasis, we employ a PyMT-driven spontaneous murine mammary carcinoma model with mammary specific deletion of *Hif1α, Hif2α*, or von Hippel-Lindau factor (*Vhl*) using the Cre-lox system. Here we show that *Hif1α* or *Hif2α* deletion in the primary tumor decreases metastatic tumor burden in the bone marrow, while *Vhl* deletion increases bone tumor burden, as hypothesized. Unexpectedly, *Hif1α* deletion increases metastatic tumor burden in the lung, while deletion of *Hif2α* or *Vhl* does not affect pulmonary metastasis. Mice with *Hif1α* deleted tumors also exhibit reduced bone volume as measured by micro computed tomography, suggesting that disruption of the osteogenic niche may be involved in the preference for lung dissemination observed in this group. Thus, we reveal that HIF signaling in breast tumors controls tumor dissemination in a site-specific manner.

[1] Graduate Program in Cancer Biology, Vanderbilt University, Nashville, TN, USA. [2] Vanderbilt Center for Bone Biology, Vanderbilt University, Nashville, TN, USA. [3] Department of Medicine, Vanderbilt University Medical Center, Nashville, TN, USA. [4] Tumor Microenvironment and Metastasis Section, Pediatric Oncology Branch, Center for Cancer Research, National Cancer Institute, National Institutes of Health, Bethesda, MD, USA. [5] Department of Medicine, Health, and Society, Vanderbilt University, Nashville, TN, USA. [6] Department of Molecular Physiology and Biophysics, Vanderbilt University, Nashville, TN, USA. [7] Department of Pathology, Microbiology and Immunology,  Vanderbilt University Medical Center, Nashville, TN, USA. [8] Department of Chemical and Biomolecular Engineering, Vanderbilt University, Nashville, TN, USA. ✉email: rachelle.johnson@vumc.org

Tumor cells frequently experience hypoxic conditions as the metabolic demands of the proliferating cells exceed the supply of oxygen and nutrients from the existing blood vessels. Cells must adapt to these stressful conditions by altering their metabolism and recruiting new blood vessels to increase oxygen supply[1]. The activation of hypoxia-inducible factor (HIF) signaling triggers these transcriptomic adaptations in response to low oxygen levels. Under normoxic conditions, HIF 1 alpha (HIF1α) and HIF2α are hydroxylated by prolyl hydroxylation domain (PHD)-containing enzymes[2–5], which allows von Hippel-Lindau factor (VHL)[6], an E3 ubiquitin ligase, to bind to and ubiquitinate HIF1α and HIF2α, marking them for proteasomal degradation[7–9]. In hypoxic conditions, PHD enzymes are inactive, leading to the stabilization of the alpha subunits, which can then dimerize with HIF1β and translocate to the nucleus[10,11]. Once in the nucleus, the HIF1 and HIF2 complexes bind to hypoxia response elements in the promoters of hypoxia-responsive genes and act as transcription factors[12–14]. HIF1α drives the expression of genes such as vascular endothelial growth factor (VEGF) to stimulate angiogenesis[15,16], as well as glucose transporters, glycolytic enzymes, and lactose dehydrogenase A to shift the main energy source of the cell to glycolysis[17–21]. HIF signaling also promotes tumor cell metastasis by driving the expression of genes that control epithelial-to-mesenchymal transition (EMT), invasion, and extracellular matrix composition[22–24]. Furthermore, clinical evidence from breast cancer patients shows that high HIF1α levels in primary tumors correlate with poor patient outcomes[25–28], and a hypoxic transcriptomic signature in tumor cells is associated with bone metastasis[29,30]. As such, HIF inhibitors are an attractive therapeutic avenue and are currently in development and undergoing clinical trials[31].

High breast cancer patient morbidity and mortality arises from metastatic dissemination of tumor cells from the primary site, which occurs early in tumor progression[32,33]. Approximately 70% of breast cancer patients present with lung or bone metastases upon autopsy[34,35]. Thus, there is an urgent need to identify the molecular factors that drive tumor dissemination to distant metastatic sites. Previous studies suggest that HIF1α expression in breast cancer cells promotes lung dissemination in genetic models[36] and bone colonization and osteolysis following intracardiac or orthotopic inoculation of MDA-MB-231 human breast cancer cells[37–39]. However, the comparative contributions of Hif1α and Hif2α on spontaneous bone dissemination are not well understood. Furthermore, dissemination patterns to bone and lung, the two leading sites of metastasis for breast cancer, have not been simultaneously evaluated in a spontaneous metastasis model to the best of our knowledge. We therefore generated three transgenic mouse models of spontaneous mouse mammary carcinoma with tumor-specific deletion of Hif1α, Hif2α, and Vhl and evaluated the effects of HIF modulation on tumor dissemination to multiple distant sites using highly sensitive detection techniques that have been optimized to detect low levels of disseminated tumor burden[40]. We found that deletion of Hif1α or Hif2α decreased dissemination to bone, while Vhl deletion increased bone dissemination. In contrast, Hif1α deletion increased lung metastatic tumor burden while Hif2α or Vhl deletion had no effect on lung dissemination. Thus, this study highlights the ability of HIF signaling to differentially modulate metastasis to certain sites.

## Results

### Deletion of Hif1α increases total tumor burden while slowing primary tumor growth.
To assess the impact of primary tumor HIF1α expression on breast tumor cell dissemination to the bone, we generated an immune-competent spontaneous murine mammary carcinoma model with mammary-specific deletion of Hif1α using the Cre-lox system. In this model, mammary tumors grow spontaneously, driven by the polyoma middle T antigen (PyMT) expressed under the mouse mammary tumor virus long terminal repeats (MMTV-LTR), which restricts the oncogene expression to the mammary epithelium. In this study, we compared female Hif1α$^{-/-}$ PyMT$^+$ mice (Hif1α$^{f/f}$, Cre-positive, PyMT-positive) to control Hif1α$^{f/f}$ PyMT$^+$ mice (Hif1α$^{f/f}$, Cre-negative, PyMT-positive) (Fig. 1a). In this spontaneous model, mammary tumors were first palpable around 8 weeks of age and reached end point (any tumor reaching 1 cm in diameter) around 20 weeks of age for the Hif1α$^{f/f}$ PyMT$^+$ mice, while the Hif1α$^{-/-}$ PyMT$^+$ mice took significantly longer to reach end point, at an average of 23 weeks of age (Fig. 1b, c). This is consistent with previous reports using this model[36]. Interestingly, the total tumor weight was greater on average in Hif1α$^{-/-}$ PyMT$^+$ mice upon sacrifice (Fig. 1d), while the number of tumors per mouse was not significantly increased (Fig. 1e).

We confirmed through real-time quantitative polymerase chain reaction (PCR) analysis of tumor homogenate RNA that Hif1α$^{-/-}$ tumors had significant deletion of Hif1α (Fig. 1f), while transcript levels of Hif2α, the other major HIF pathway signaling factor, was unaffected (Fig. 1g). Recombination and deletion of Hif1α was further confirmed by PCR amplification of the Hif1α locus (Supplementary Fig. 1a, b). Hif1α and Hif2α transcript levels were unaffected in the lung, suggesting that we did not have off-target editing in other soft tissue sites (Supplementary Fig. 1c, d). Furthermore, the expression of Vegfa, a downstream target of Hif1α, was significantly lower in Hif1α$^{-/-}$ tumors (Fig. 1h), while the staining intensity of pimonidazole, a hypoxia marker, was comparable in Hif1α$^{f/f}$ and Hif1α$^{-/-}$ tumors (Fig. 1i, j). This confirms that Hif1α$^{-/-}$ tumors have a dampened hypoxia response despite experiencing similar levels of hypoxia as Hif1α$^{f/f}$ tumors.

### Deletion of Hif1α in the mammary fat pad reduces trabecular bone volume.
Since bone disseminated breast tumor cells can cause the formation of osteolytic lesions, we first assessed trabecular bone volume of tibiae from Hif1α$^{f/f}$ PyMT$^+$ and Hif1α$^{-/-}$ PyMT$^+$ mice by microcomputed tomography (microCT) as a readout of tumor-induced bone destruction. To ensure that Hif1α deletion in the mammary fat pad did not alter baseline bone microarchitecture, we also scanned tibiae from non-tumor bearing controls (PyMT$^-$) of each genotype. Bone volume and trabecular number were significantly decreased in Hif1α$^{-/-}$ PyMT$^+$ mice, with an accompanying increase in trabecular spacing (Fig. 2a–d), but bone volume was unaltered with Hif1α deletion in PyMT$^-$ mice, suggesting that the reduction in bone volume observed in Hif1α$^{-/-}$ PyMT$^+$ mice was due to tumor-induced osteolysis. Hif1α$^{-/-}$ PyMT$^-$ mice did exhibit a significant increase in trabecular thickness (Fig. 2e), but this may not be biologically significant given the absence of changes in any other structural parameters for bone. Surprisingly, we found no discernible tumor burden present in the bone marrow of Hif1α$^{-/-}$ PyMT$^+$ mice upon histological inspection by a certified veterinary pathologist, save in one mouse (Fig. 2f, g). Thus, Hif1a deletion in the primary tumor decreases trabecular bone volume but does not drive the development of macroscopic or osteolytic lesions in the bone.

### Deletion of Hif1α decreases bone dissemination while increasing lung metastasis.
Given the lack of macroscopic tumor lesions in the bone and the fact that the main site of metastasis for the PyMT-driven mammary carcinoma model is the lung rather than the bone marrow[41,42], we employed several sensitive

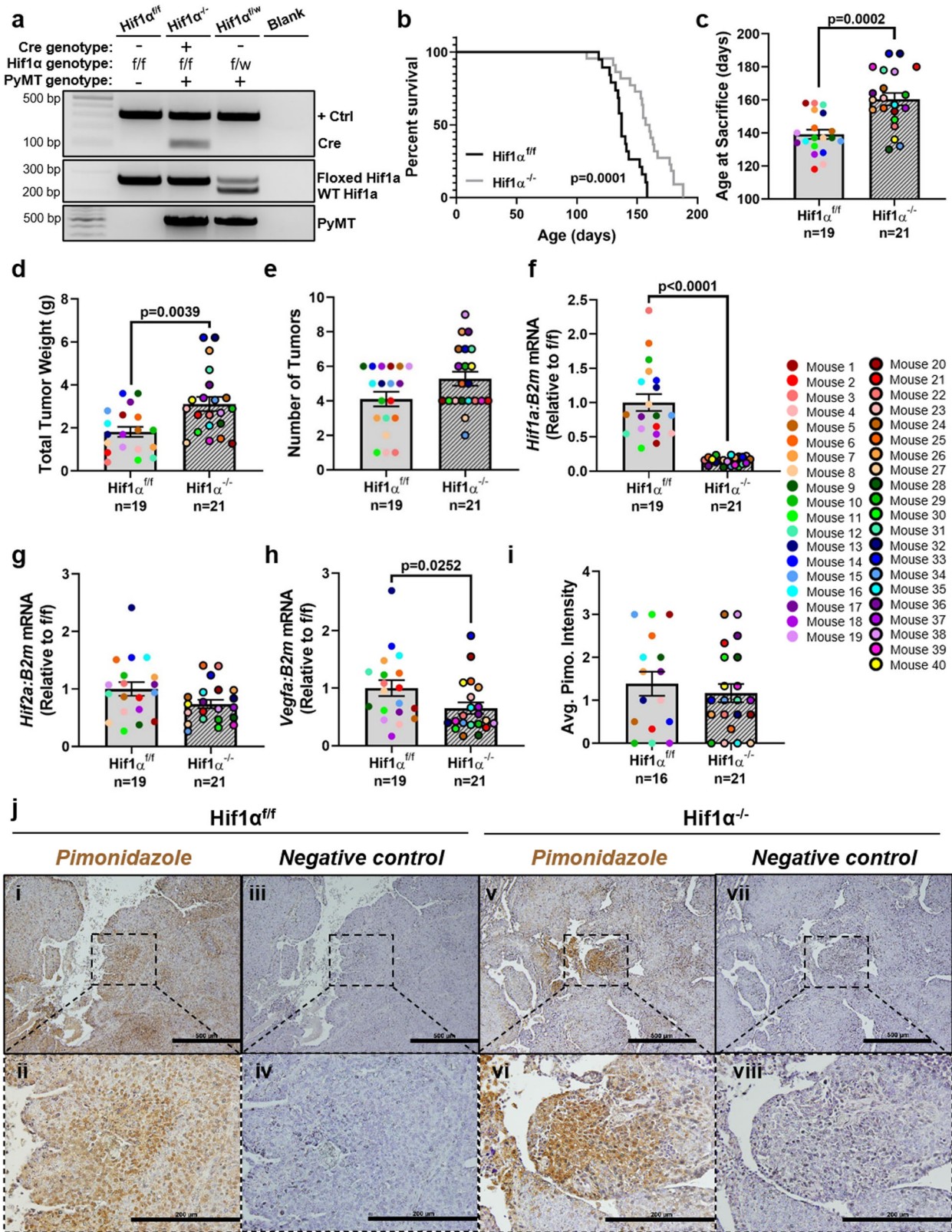

strategies to determine whether there were differences in tumor dissemination to bone. First, tumor cells were detected by flow cytometry of bone marrow from the tibiae and femora from Hif1α$^{f/f}$ PyMT$^+$ and Hif1α$^{-/-}$ PyMT$^+$ mice based on epithelial cell adhesion molecule (EpCAM) positivity. While some background EpCAM staining was detected in non-tumor-bearing

(PyMT$^-$) bone marrow (Supplementary Fig. 2), a significant decrease in the number of EpCAM$^+$ cells was detected in the bone marrow from Hif1α$^{-/-}$ PyMT$^+$ mice compared to Hif1α$^{f/f}$ PyMT$^+$ mice when normalized to total tumor weight at end point (Fig. 3a and Supplementary Fig. 3a). Second, the tumor-specific *Pymt* transcript was quantified from femur and spine

**Fig. 1 Deletion of *Hif1α* increases total tumor burden while slowing primary tumor growth. a** Example genotyping gel of Cre, floxed Hif1α, and PyMT from Hif1α[f/f] PyMT[+], Hif1α[−/−] PyMT[+], and Hif1α[f/w] PyMT[−] mice. "+ Ctrl" product indicates an internal positive control. **b** Survival analysis of Hif1α[f/f] PyMT[+] and Hif1α[−/−] PyMT[+] mice where end point represents sacrifice due to tumor size reaching collection threshold. Log-rank test. **c–e** Comparison of the age at sacrifice, total tumor burden at sacrifice, and number of tumors collected per mouse. Two-tailed Mann–Whitney test. **f–h** Quantitative PCR analysis of *Hif1α*, *Hif2α*, and *Vegfa* expression compared to *B2m* from whole-tumor homogenate RNA. Expression is normalized to the mean of the f/f control group. Two-tailed Mann–Whitney test. **i** The average pimonidazole staining intensity for each mouse across multiple images from a single tumor. Two-tailed Mann–Whitney test. **j** Representative images of pimonidazole staining, taken with ×10 and ×40 objectives. The ×40 field is denoted with a dashed box in the ×10 view. Scale bars represent 500 μm in ×10 fields, and 200 μm in ×40 fields. Negative controls lacked the primary antibody incubation step. Graphs represent mean per group and error bars represent s.e.m. *n* = 19 (or 16 in **i**) Hif1α[f/f] PyMT[+] mice and *n* = 21 Hif1α[−/−] PyMT[+] mice.

homogenate RNA as a marker of tumor burden (Fig. 3b, c and Supplementary Fig. 3b, c). No significant difference in tumor burden was detected in the femur and spine of Hif1α[−/−] PyMT[+] mice using this method. Taken together, these results suggest that *Hif1α* in the primary tumor promotes tumor cell dissemination to the bone.

In addition to quantifying tumor burden in bone sites, we quantified tumor burden in the lung by histological inspection in order to confirm that Hif1α[−/−] PyMT[+] mice had decreased tumor burden, as reported previously[36]. Surprisingly, Hif1α[−/−] PyMT[+] mice had a significant increase in the number of metastatic lesions in the lung (Fig. 3d and Supplementary Fig. 3d, e) and total metastatic lesion area (Fig. 3e and Supplementary Fig. 3f, g), regardless of whether the data were normalized to total tumor weight at end point, the mouse age at sacrifice, or left un-normalized. Individual lesion size was not significantly different between the two groups (Supplementary Fig. 3h), indicating that the increase in total tumor area is driven by the greater number of lesions. Hif1α[−/−] PyMT[+] mice also had significantly greater incidence of pulmonary lesions compared to Hif1α[f/f] PyMT[+] mice (Fig. 3f). A non-significant trend in increased tumor burden was observed in the contralateral lung of Hif1α[−/−] PyMT[+] mice by *Pymt* transcript levels (Supplementary Fig. 3i, j). To determine whether the increased tumor burden in the lung was indicative of a global metastatic increase in soft tissue sites, tumor burden was also measured in the brain and liver. There was a non-significant trend toward a reduction in tumor burden in the brain by *Pymt* transcript levels in Hif1α[−/−] PyMT[+] mice (Supplementary Fig. 3k, l), and no liver lesions were detected by histological analysis by a certified veterinary pathologist (Supplementary Fig. 3m). Thus, *Hif1a* deletion in the primary tumor reduces bone dissemination but also specifically promotes lung metastasis.

There was no difference in the expression of general breast cancer metastasis-associated genes (Fig. 4a), nor in EMT markers (Fig. 4b), between primary tumors from the Hif1α[f/f] PyMT[+] and Hif1α[−/−] PyMT[+] mice. This is consistent with site-specific differences in tumor burden, rather than a global increase or decrease in dissemination to all the distant sites measured. We also measured the expression of a panel of autophagy markers, since previous reports have shown that hypoxia-induced autophagy can affect pulmonary tumor burden. Inhibition of autophagy in hypoxic breast cancer cells has been shown to promote pulmonary metastasis[43], while autophagy has also been shown to promote the survival of dormant breast cancer cells and promote metastatic tumor recurrence[44]. However, we saw no differences between Hif1α[f/f] and Hif1α[−/−] tumors in any of the autophagy markers tested (Fig. 4c).

We therefore examined gene expression data from The Cancer Genome Atlas (TCGA) Invasive Breast Carcinoma patient dataset to determine whether hypoxia differentially induced gene expression profiles associated with site-specific metastasis to the bone or lung. We found that a 42-gene hypoxia signature identified by Ye et al.[45] significantly and positively correlated with the bone metastasis signature established by Kang et al.[46]

(Fig. 4d). Thus, hypoxia appears to drive a large transcriptomic program that promotes bone dissemination. However, a similar trend was observed for the lung metastasis signature from Minn et al.[47] (Fig. 4e). Taken together, it appears that hypoxia upregulates tumor dissemination in general at the transcriptomic level, suggesting that the increase in lung tumor burden may be due to altered Hif1α[−/−] tumor cell response to signals from the lung microenvironment.

**Proliferation and immune markers are unaltered in lung metastatic Hif1α[−/−] lesions.** To reconcile the divergent role of *Hif1α* in tumor cell dissemination to bone and lung, we therefore sought to identify factors that may specifically be driving outgrowth of tumor cells in the lung. Since there is evidence that hypoxia promotes tumor dormancy specifically in the lung[48], we investigated whether the increased lung tumor burden in Hif1α[−/−] PyMT[+] mice was due to dormancy escape by the lung-disseminated tumor cells. First, we confirmed that there was no significant difference in pimonidazole-positive lung lesion area, indicating that Hif1α[f/f] and Hif1α[−/−] tumor cells experience similar levels of hypoxia in the lung (Supplementary Fig. 4a, b). There was, however, no difference in Ki-67[+] lung lesion area between the groups, indicating that *Hif1α* expression does not alter the pro-liferative capacity of the disseminated cells once they exit dormancy (Fig. 4f, g). We next measured the abundance of CD4[+] and CD8[+] cells in the lungs, since altered immune cell numbers in the lung tissue may indicate differences in immune-mediated tumor cell clearance. However, there was no difference in CD4[+] or CD8[+] T cell infiltration in the lungs of Hif1α[f/f] PyMT[+] and Hif1α[−/−] PyMT[+] mice (Fig. 4h, i). Immune cell numbers were not significantly different between tumor-bearing lungs of Hif1α[f/f] PyMT[+] and Hif1α[−/−] PyMT[+] mice either (red points, Fig. 4h). To investigate whether pre-metastatic niche development could be driving the outgrowth of lung-disseminated tumor cells, we measured the expression of lysyl oxidase (LOX), a hypoxia-dependent tumor-secreted factor known to drive pre-metastatic niche development[49]. However, there was no change in *Lox* expression between Hif1α[f/f] and Hif1α[−/−] tumors (Fig. 4j).

**Deletion of *Hif2α* decreases tumor dissemination to the bone but not to the lung.** While Hif1α and Hif2α have some redundant functions, they have unique and sometimes opposing effects[50]. To determine whether the dissemination patterns observed in the Hif1α[−/−] PyMT[+] mice were due to a general decrease in HIF signaling activity, or due to Hif1α-specific effects, we generated Hif2α[f/f] PyMT[+] and Hif2α[−/−] PyMT[+] mice using the same breeding strategy as the Hif1α[−/−] PyMT[+] mice. We observed no difference between the Hif2α[f/f] PyMT[+] and Hif2α[−/−] PyMT[+] mice in the time it took tumors to reach collection size (Supplementary Fig. 5a, b), the total tumor weight upon sacrifice (Supplementary Fig. 5c), or the number of tumors (Supplementary Fig. 5d). We confirmed a significant reduction in *Hif2α* transcript levels from whole-tumor homogenate RNA of Hif2α[−/−] PyMT[+]

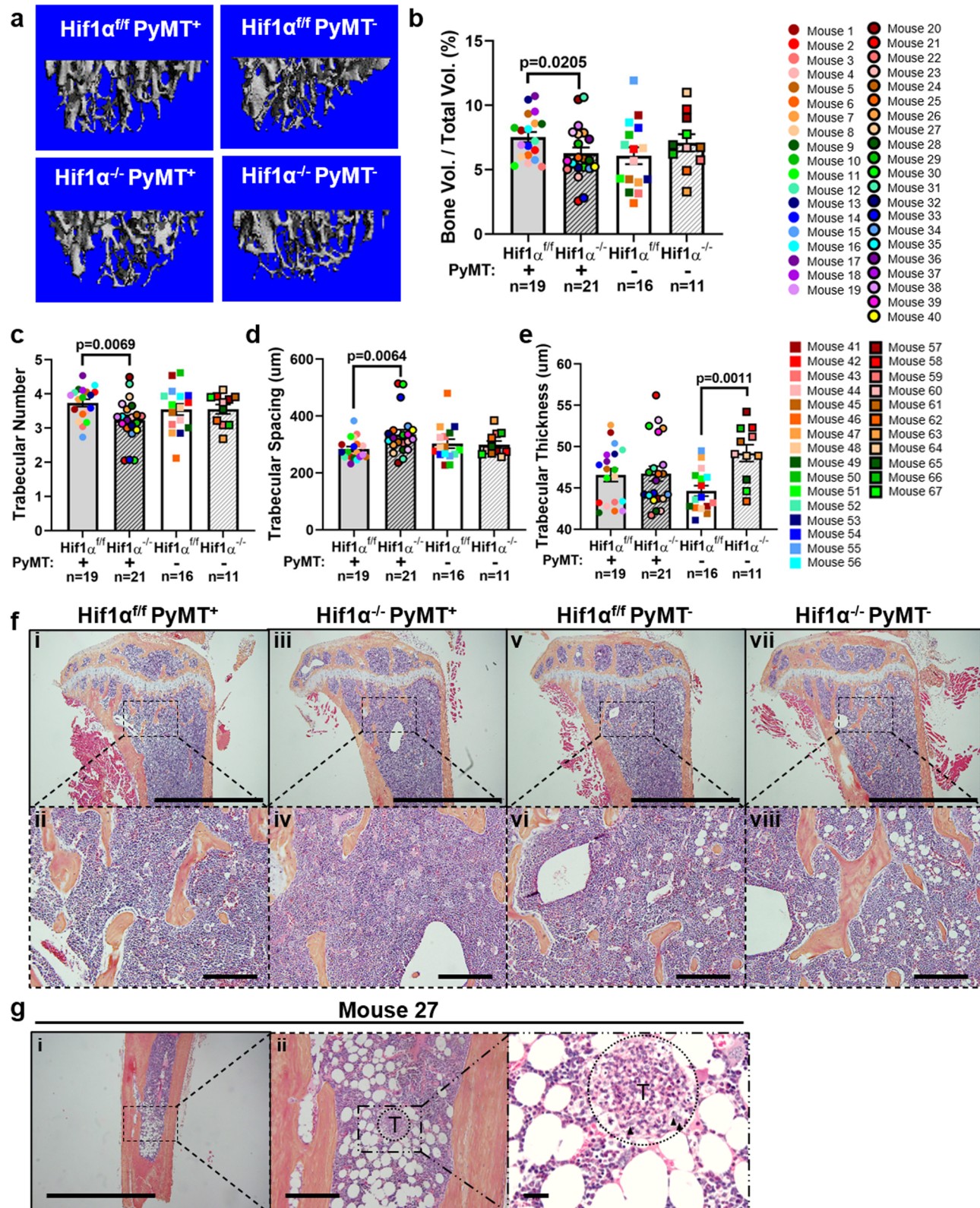

mice (Supplementary Fig. 5e) and no difference in *Hif1α* transcript levels (Supplementary Fig. 5f). *Hif2α* deletion did not significantly reduce *Vegfa* expression in the primary tumor (Supplementary Fig. 5g).

While microCT analysis of Hif2α$^{-/-}$ PyMT$^+$ mice did not reveal any differences in trabecular bone parameters (Supplementary Fig. 5h–l), flow cytometric analysis of hindlimb bone

marrow from Hif2α$^{-/-}$ PyMT$^+$ mice revealed a significant reduction in the percentage of EpCAM$^+$ tumor cells (Fig. 5a). However, there was no significant difference in the *Pymt* transcript abundance in the femur or spine (Fig. 5b, c). Unlike the Hif1α$^{-/-}$ PyMT$^+$ mice, lung metastatic tumor burden in Hif2α$^{-/-}$ PyMT$^+$ mice was not significantly different than Hif2α$^{f/f}$ PyMT$^+$ mice, as measured by lesion number, total lesion

**Fig. 2 Deletion of *Hif1α* in the mammary fat pad reduces trabecular bone volume. a** Representative 3D renderings of microCT scans of the proximal metaphysis of the right tibia. **b–e** Quantification of bone volume as a percentage of total volume, trabecular number, trabecular spacing, and trabecular thickness. Two-tailed Mann–Whitney test against corresponding f/f control. **f** Representative H&E-stained images of the proximal metaphysis of the tibia taken with ×4 and ×20 objectives. The ×20 field is denoted with a dashed box in the ×4 view. Scale bars represent 2.0 mm in ×4 view and 200 μm in the ×20 view. **g** H&E-stained images of the one histologically detectable bone metastatic lesion, identified in the distal region of the tibia from Mouse 27. Lower-magnification images taken with ×4 (i), ×20 (ii) objectives, while highest magnification image (iii) is taken at ×600 magnification. Scale bars represent 2.0 mm in ×4 view, 200 μm in the ×20 view, and 25 μm in ×600 view. Area of higher magnification is indicated in lower magnification images with a dashed box. Tumor area is labeled T and indicated by dashed circle. Atypical cells denoted with arrowheads. Graphs represent mean per group and error bars represent s.e.m. $n = 19$ Hif1α$^{f/f}$ PyMT$^+$ mouse tibiae, $n = 21$ Hif1α$^{-/-}$ PyMT$^+$ mouse tibiae, $n = 16$ Hif1α$^{f/f}$ PyMT$^-$ mouse tibiae, and $n = 11$ Hif1α$^{-/-}$ PyMT$^-$ mouse tibiae.

area, individual lesion area, or incidence of lung metastases (Fig. 5d–g). Similarly, there was no significant difference in *Pymt* transcript abundance in the lungs or brains of Hif2α$^{-/-}$ PyMT$^+$ mice (Fig. 5h, i). Metastatic tumor burden data for Hif2α$^{-/-}$ PyMT$^+$ mice were un-normalized since there was no significant difference in tumor weight or age at sacrifice. Taken together, these results suggest that *Hif2α* in the primary tumor drives tumor cell dissemination to the bone but is dispensable for dissemination to the lung.

**Deletion of *Vhl* increases tumor dissemination to the bone but not to the lung.** To determine the effect of Hif1α and Hif2α activation, rather than deletion, we generated a third transgenic mouse strain in which we could model constitutive HIF signaling activation through deletion of Vhl, a negative regulator of HIF signaling. Vhl$^{f/f}$ PyMT$^+$ and Vhl$^{-/-}$ PyMT$^+$ mice were generated using the same strategy as the other transgenic strains. Vhl$^{-/-}$ tumors took a significantly longer time to reach collection size compared to Vhl$^{f/f}$ controls (Fig. 6a, b) but had a significant reduction in total tumor weight upon sacrifice (Fig. 6c). The average number of tumors was not significantly different between Vhl$^{f/f}$ PyMT$^+$ and Vhl$^{-/-}$ PyMT$^+$ mice (Fig. 6d). While we were unable to detect a decrease in *Vhl* transcript in whole tumor RNA from Vhl$^{-/-}$ PyMT$^+$ mice (Fig. 6e), we confirmed recombination of the *Vhl* locus at the genomic level (Fig. 6f) and observed a significant increase in the expression of *Vegfa*, phosphoglycerate kinase 1 (*Pgk1*), glucose transporter type 1 (*Glut1*), TEK receptor tyrosine kinase (*Tek*, also known as Tie2), and platelet-derived growth factor subunit B (*Pdgfb*) (Fig. 6g–k), indicating that HIF downstream signaling is elevated with *Vhl* deletion.

Despite unaltered trabecular bone parameters (Fig. 7a, b and Supplementary Fig. 7a–c), *Vhl* deletion increased the number of tumor cells detected in the bone marrow by flow cytometry when normalized to tumor weight (Fig. 7c and Supplementary Fig. 6d). No significant differences were observed in the *Pymt* transcript abundance in the femur or spine (Fig. 7d, e and Supplementary Fig. 6e, f). From histological analysis of the lung, no significant differences were observed in total metastatic lesion number or area between Vhl$^{f/f}$ PyMT$^+$ and Vhl$^{-/-}$ PyMT$^+$ mice, regardless of normalization to total primary tumor weight (Fig. 7f, g and Supplementary Fig. 6g, h). In addition, the incidence of lung metastases was no different between the two groups (Fig. 7h). There was a slight increase in the average individual lung lesion size in the Vhl$^{-/-}$ PyMT$^+$ mice (Supplementary Fig. 6i), but this difference was not sufficient to drive an increase in the overall lung tumor burden. Additionally, there were no significant differences in *Pymt* transcript abundance in the lungs or brains of Vhl$^{-/-}$ PyMT$^+$ mice (Fig. 7i, j and Supplementary Fig. 6j, k).

While *Hif1α* deletion was not found to alter the expression of genes involved in specific metastasis-related signaling pathways (Fig. 4), *Vhl* deletion was found to increase, although not significantly, the expression of *Pthlh* in the primary tumor (Fig. 8a). *PTHLH* and its gene product PTHrP are known to

promote breast tumor progression and metastasis[51] and are key drivers of tumor-induced bone disease[52–54]. Additionally, *PTHLH* is a direct target of HIF2α[55]. Accordingly, we observed that *Hif2α*, but not *Hif1α*, deletion drove a modest, non-significant decrease in *Pthlh* expression (Fig. 8b, c). *Vhl* deletion also modestly increased the expression of *Cxcr4* (Fig. 8d), a gene known to drive breast cancer cell metastasis to bone[56,57], though this increase did not reach the threshold of statistical significance. *Hif1α* deletion did not result in a reciprocal decrease in *Cxcr4* expression, but *Hif2α* deletion yielded a non-significant decrease in *Cxcr4* (Fig. 8e, f). Due to the subtlety of expression changes observed for *Pthlh* and *Cxcr4*, other HIF-driven pro-metastatic factors are most likely involved in the mechanism behind HIF-driven bone metastasis (Fig. 8g).

## Discussion

While our main analysis was focused on dissemination patterns and quantifying tumor burden in various distant sites, we observed surprising patterns in primary tumor growth. The slower tumor progression we observed in the Hif1α$^{-/-}$ tumors has been reported previously in this model[36] and has been attributed to delayed microvessel growth in the mammary fat pad due to insufficient HIF signaling. Interestingly, while tumor progression is delayed, total tumor weight was higher on average in Hif1α$^{-/-}$ PyMT$^+$ mice than in Hif1α$^{f/f}$ PyMT$^+$ mice. However, in the previous study using this model[36], no significant difference in tumor weight at end point was noted. In contrast to *Hif1α*, our data indicate that *Hif2α* deletion does not significantly alter tumor progression, suggesting that HIF1α and HIF2α target distinct pathways that regulate tumor growth. Interestingly, while we expected *Vhl* deletion to yield the opposite tumor growth pattern from *Hif1α* deletion and cause tumors to grow more rapidly, *Vhl*-deleted tumors in fact progressed more slowly. VHL deletion has been reported to increase immune infiltration in mammary tumors[58], which may contribute to increased tumor cell clearance and the slower tumor growth and decreased total tumor weight observed in the Vhl$^{-/-}$ PyMT$^+$ mice.

Our finding that bone metastasis was significantly lower with *Hif1α* or *Hif2α* deletion are highly consistent with previous studies demonstrating that hypoxic signaling promotes tumor cell dissemination to the bone. HIF1α expression in breast cancer cells promotes bone colonization and osteolysis following intracardiac or orthotopic inoculation of MDA-MB-231 human breast cancer cells[37–39], and hypoxic transcriptomic signatures in breast cancer cells has been associated with bone metastasis[29,30].

It is important to note that our lung metastasis findings, which indicate that Hif1α inhibits lung metastasis while Hif2α has no effect, contradict a large body of work which demonstrates that both Hif1α and Hif2α promote lung metastasis. A prior study using the genetic MMTV-PyMT mouse model with mammary-specific deletion of Hif1α indicated that Hif1α expression promoted lung metastasis[36]. The reason for the differences in our findings from previous studies are likely multifaceted. We used

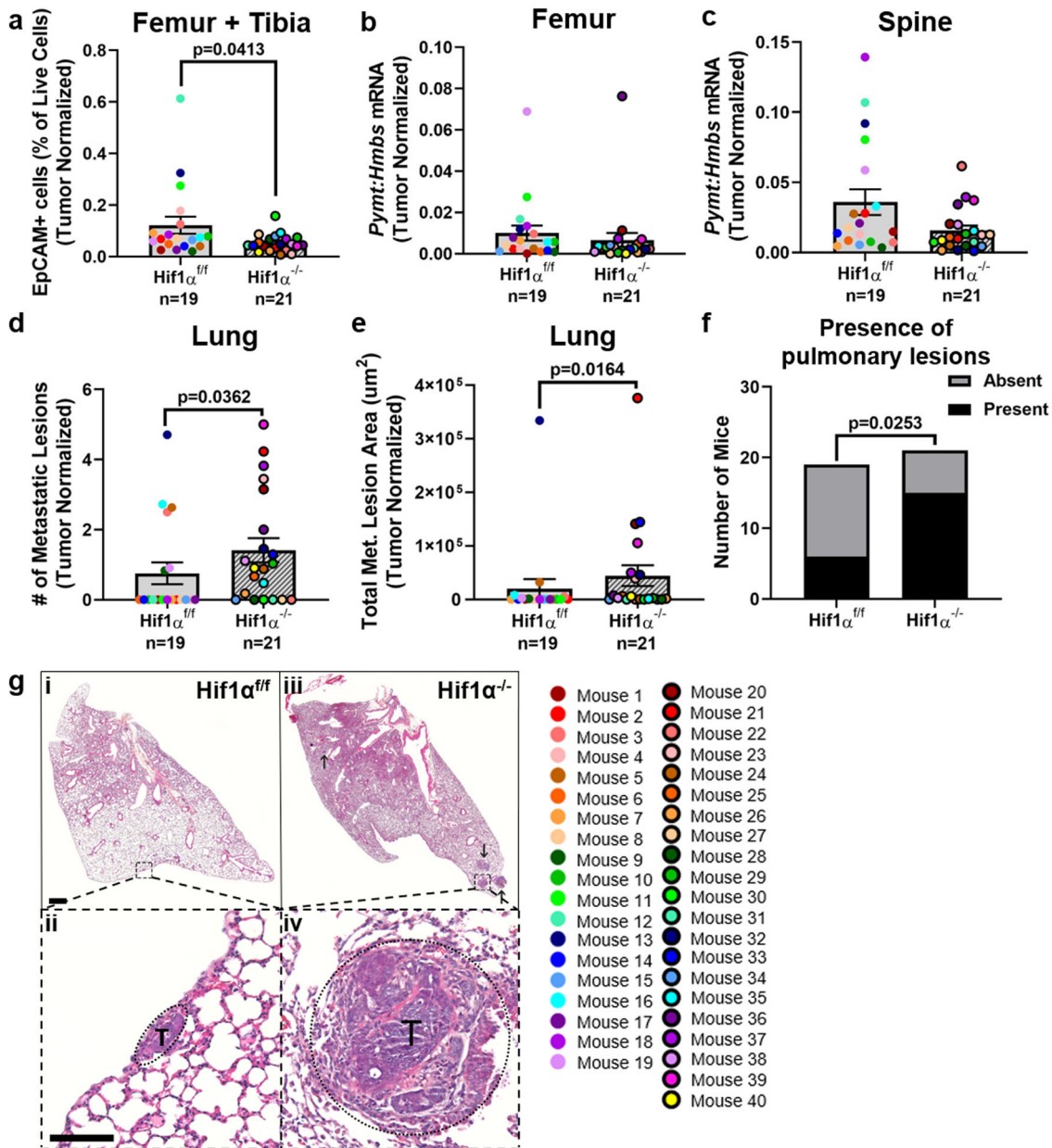

**Fig. 3 Deletion of *Hif1α* decreases bone dissemination while increasing lung metastasis. a** The percentage of EpCAM[+] cells, out of the total number of live cells, detected by flow cytometric analysis of left hindlimb bone marrow. Numbers are normalized to the total tumor weight of each mouse. Two-tailed Mann–Whitney test. **b**, **c** Quantitative PCR analysis of *Pymt* transcript compared to *Hmbs* from right femur or spinal midsection, respectively. *Pymt* expression was then normalized to the total tumor weight of each mouse. Two-tailed Mann–Whitney test. **d**, **e** Metastatic lesion number or area detected by histological analysis of H&E-stained sections from the left lung. Numbers are normalized to the total tumor weight of each mouse. Two-tailed Mann–Whitney test. **f** Comparison of the proportion of mice from each group that had any detectable pulmonary lesions. Fisher's exact test. **g** Representative images of H&E-stained lung sections. Scale bar in sub-gross photomicrographs is 500 μm. Scale bar in high-power micrographs is 100 μm. The high-power image field is denoted with a dashed box in the sub-gross view. Tumor area is denoted with a dashed oval in the high-power images (labeled T). Additional metastatic lesions in the sub-gross view are denoted with arrows. Graphs represent mean per group and error bars represent s.e.m. *n* = 19 Hif1α[f/f] PyMT[+] mice, *n* = 21 Hif1α[−/−] PyMT[+] mice.

similar methodology, and we had a board-certified, blinded veterinary pathologist perform the lung tumor analysis. It is possible that the differences may be due to differences in genetic strain, since the combination of these mice is on a mixed genetic background. Our colony is therefore genetically distinct from the previous study. Significant differences have also been noted in microbiome composition of genetically engineered mice with the same genotype housed at different facilities[59]. The commensal microbiota can significantly impact both local (gut) and systemic

immunity[60,61] and thus could also influence the clearance or growth of disseminated tumor cells. Minor variations in animal handling and housing conditions, such as cage density and diet, have also been shown to affect biochemical, hematological, metabolic, and endocrine parameters[62].

Additional studies similarly indicate that increased HIF signaling increases metastasis, including a 45-gene hypoxia response gene signature that was predictive of breast cancer patients' risk of developing lung metastases[37]. Several studies

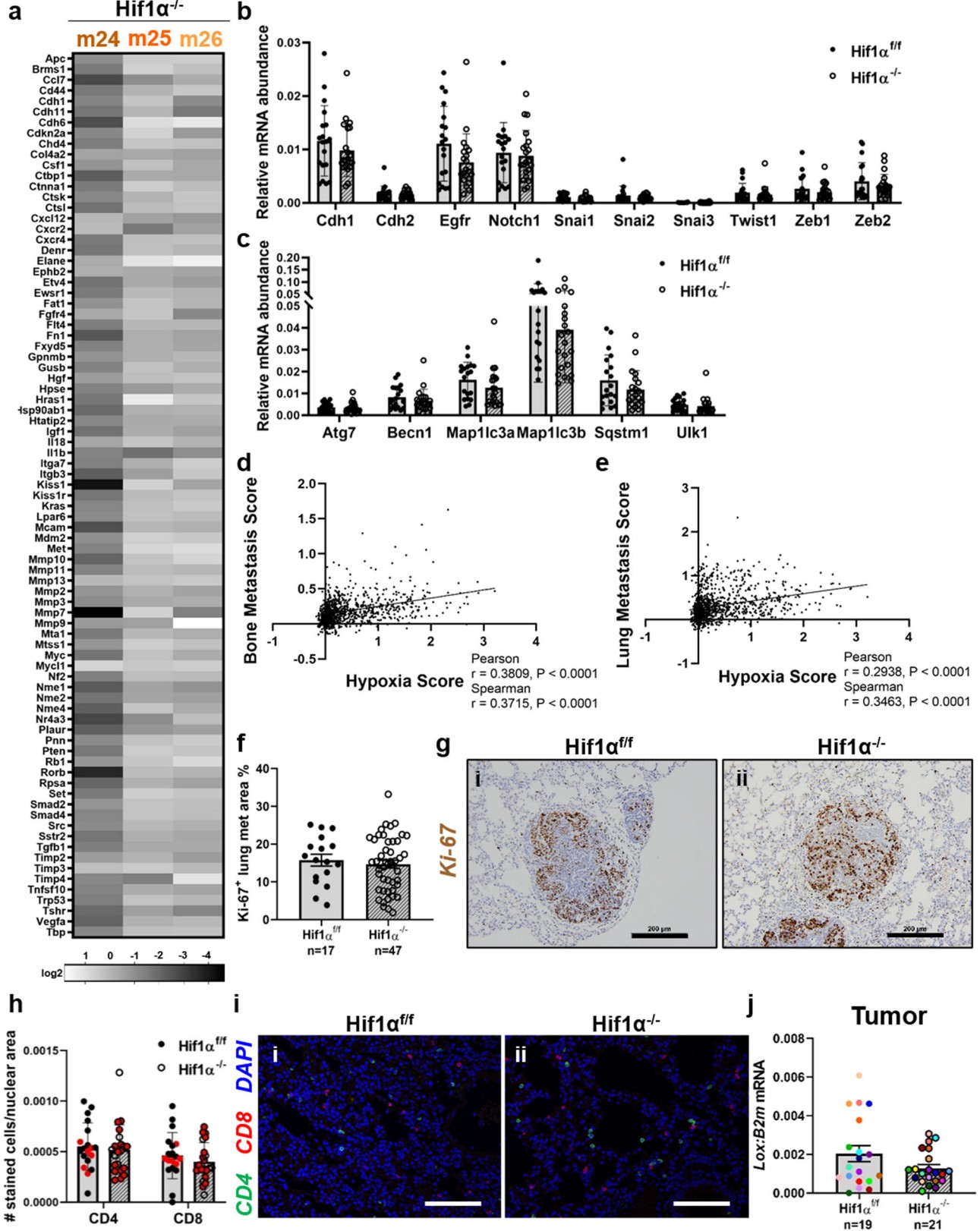

using MDA-MB-231 human breast cancer cells or EMT6 murine mammary carcinoma cells have also demonstrated that inhibition of HIF1α or HIF2α significantly diminishes metastasis to the lung[37,63,64]. HIF signaling has been shown to facilitate tumor cell extravasation by driving the expression of genes that increase breast cancer cell adhesion to endothelial cells, such as L1 cell adhesion molecule, as well as genes that decrease adhesion between endothelial cells, such as angiopoietin-like 4[63]. Specific downstream effectors of HIF signaling that prepare the lung microenvironment for metastatic colonization, such as LOX and LOX-like proteins (LOXL2 and LOXL4), have also been identified[23,65]. A potential

**Fig. 4 Hypoxia correlates with pro-metastatic transcription signatures. a** Heatmap depiction of the expression of a panel of metastasis-associated genes from $n = 3$ Hif1α$^{-/-}$ tumors compared to the average expression of $n = 3$ Hif1α$^{f/f}$ tumors. The difference in expression of each gene between the three Hif1α$^{f/f}$ and three Hif1α$^{-/-}$ tumors was compared using the Mann–Whitney test. **b** Expression of a panel of EMT-related genes (*Cdh1, Cdh2, Egfr, Notch1, Snai1, Snai2, Snai3, Twist1, Zeb1, Zeb2*) compared to *B2m*, measured by qPCR. Two-tailed Mann–Whitney test against corresponding f/f control. **c** Expression of a panel of autophagy-related genes (*Atg7, Becn1, Map1lc3a, Map1lc3b, Sqstm1, Ulk1*) compared to *B2m*, measured by qPCR. Two-tailed Mann–Whitney test against corresponding f/f control. **d, e** Correlation of Ye et al. hypoxia gene signature[45] with mRNA levels of genes in the Kang et al. bone metastasis signature[46] or the Minn et al. lung metastasis signature[47], respectively, in TCGA Invasive Breast Carcinoma patient dataset. Pearson and Spearman correlation. $n = 1100$ patients. **f** Quantification of Ki-67 area as a percentage of the lesion area. Two-tailed Mann–Whitney test. $n = 17$ individual Hif1α$^{f/f}$ metastatic lung lesions, $n = 47$ individual Hif1α$^{-/-}$ metastatic lung lesions. **g** Representative images of Ki-67-stained lung metastatic lesions, taken with ×20 objective. Scale bars represent 200 µm. **h** Quantification of the average number of CD4$^+$ and CD8$^+$ cells in the lung normalized to the total nuclear area for the image. Data points in red denote mice that had detectable lung tumor burden. Two-tailed Mann–Whitney test against corresponding f/f control. **i** Representative images of immunofluorescent staining of CD4 and CD8, taken with ×20 objective. Scale bars represent 200 µm. **j** Quantitative PCR analysis of *Lox* transcript compared to *B2m* from primary tumor homogenate RNA. Two-tailed Mann–Whitney test. Graph represents mean per group and error bars represent s.e.m. $n = 19$ Hif1α$^{f/f}$ PyMT$^+$ mice, $n = 21$ Hif1α$^{-/-}$ PyMT$^+$ mice.

explanation for the difference in findings from these prior studies may be due to our use of a spontaneous tumor formation and dissemination model since the majority of the previous studies utilized xenograft or syngeneic injection models, or may be due to differences between PyMT tumor cells and the tumor lines utilized in the other studies. It is possible that, since the PyMT-driven spontaneous mammary cancer model has such a robust lung metastasis phenotype, contributions of HIF signaling may be negligible. This would explain why we do not see changes in lung tumor burden in the Hif2α- or Vhl-deletion models and could point to a yet unidentified mechanism at play in the Hif1α-deletion strain that exhibited increased lung metastasis.

Thus far, we have not been able to identify a molecular driver for the increased lung metastasis observed with *Hif1α* deletion and have eliminated a significant number of potential mechanisms. We demonstrated that *Hif1α* expression does not alter the proliferative capacity of macroscopic lung lesions, but due to the collection time point we were not able to detect single disseminated tumor cells in these mice. As we collected mice relatively late in disease progression in order to maximize the amount of disseminated cells, the lung lesions were already well developed. Thus, we could not determine whether the proliferative capacity of these disseminated tumor cells was altered by *Hif1α* expression prior to their development into histologically detectable lesions. We cannot rule out that disseminated tumor cells that had not yet grown into macroscopic lesions are detectable in mice collected earlier in disease progression and there may be differences in Ki-67 staining at that time point. Given the subtlety of our model and the large mouse numbers necessary for robust statistical analysis at earlier time points, it would be difficult to assess these differences earlier in the model. Importantly, normalizing lung tumor burden to age at sacrifice did not change our finding that *Hif1α* deletion increased lung metastasis, indicating that our data is likely not an artifact of the difference in collection age.

Additionally, this difference in outgrowth of disseminated tumor cells could be due to tumor cell-intrinsic properties driven by *Hif1α* expression or due to microenvironmental changes in the lung that result from *Hif1α* expression in the primary tumor. First, alterations in immune cell populations in the lung may alter the outgrowth of lung metastatic cells due to immune-mediated tumor cell clearance[66]. We measured the presence of CD4$^+$ and CD8$^+$ cells in the lungs of Hif1α$^{f/f}$ PyMT$^+$ and Hif1α$^{-/-}$ PyMT$^+$ mice but observed no differences. However, other immune cell types are also involved in tumor cell clearance. Modulating the activity or polarization of immune cells is also known to create microenvironments that oppose or support tumor growth. For example, T cell inactivation is known to create a permissive microenvironment for tumor growth[67], and tumor-associated

macrophages can promote tumor progression through many pathways, such as inducing tissue remodeling and fibrosis, taming of adaptive immunity, and providing protective niches for cancer stem cells[68]. Some immune cell types, especially CD11b$^+$ cells, have been implicated in the establishment of the pre-metastatic niche[49,69] that fosters tumor cell growth. Thus, a more extensive immune cell profiling of the lung may reveal differences that would explain the difference in tumor cell outgrowth. Second, primary tumor-secreted factors have been shown to affect the establishment of a pre-metastatic niche. Hypoxia-induced LOX secretion from breast cancer cells is a driver of pre-metastatic niche development in the lung[49] as well as osteolytic lesion development in the bone[29,70]. While we did not observe a difference in *Lox* expression that would explain the increased lung tumor burden and decreased bone volume observed in Hif1α$^{-/-}$ PyMT$^+$ mice, this is not the only factor that regulates pre-metastatic niche development. In prostate cancer, exosomes released from hypoxic cancer cells have been shown to promote matrix metalloproteinase activity at pre-metastatic sites[71]. The mechanisms that govern pre-metastatic niche development at various anatomical sites is not yet well characterized, and the impact of hypoxia on this process warrants further investigation.

Interestingly, a change in lung tumor burden was only observed in the *Hif1α*-deleted mice, and not in the *Hif2α*- or *Vhl*-deleted mice. Of note, a decrease in tibial bone volume was observed in the Hif1α$^{-/-}$ PyMT$^+$ mice, but not in the *Hif2α*- or *Vhl*-deleted mice. The osteogenic niche is key to the establishment and growth of disseminated tumor cells in the bone marrow, as the interaction of tumor cells with osteoblast lineage cells through heterotypic adherens junctions activates pro-proliferation pathways[72]. However, osteoblast-secreted factors have also been shown to induce dormancy and quiescence in prostate cancer cells[73,74], indicating that the osteogenic niche can promote quiescence of tumor cells that engraft in these regions of the bone. Since the osteogenic niche is reduced in the Hif1α$^{-/-}$ PyMT$^+$ mice, as indicated by the reduction in bone volume, the shrinkage of suitable niches available for tumor cell engraftment in bone may cause a larger portion of the metastatic tumor cells to engraft in the lung instead. This may explain why we do not observe a difference in lung tumor burden in *Hif2α*- or *Vhl*-deleted mice. The mechanism behind the decreased bone volume in Hif1α$^{-/-}$ PyMT$^+$ mice remains unclear; however, it is important to note that there was no difference in bone volume between Hif1α$^{f/f}$ and Hif1α$^{-/-}$ PyMT$^-$ mice, suggesting that the reduction in bone volume is not due to deletion of *Hif1α* in off-target cell lineages, but rather due to tumor-related factors.

While a loss of the osteogenic niche could explain the reduction in bone dissemination and increase in lung metastasis observed in the Hif1α$^{-/-}$ PyMT$^+$ mice, other mechanisms must

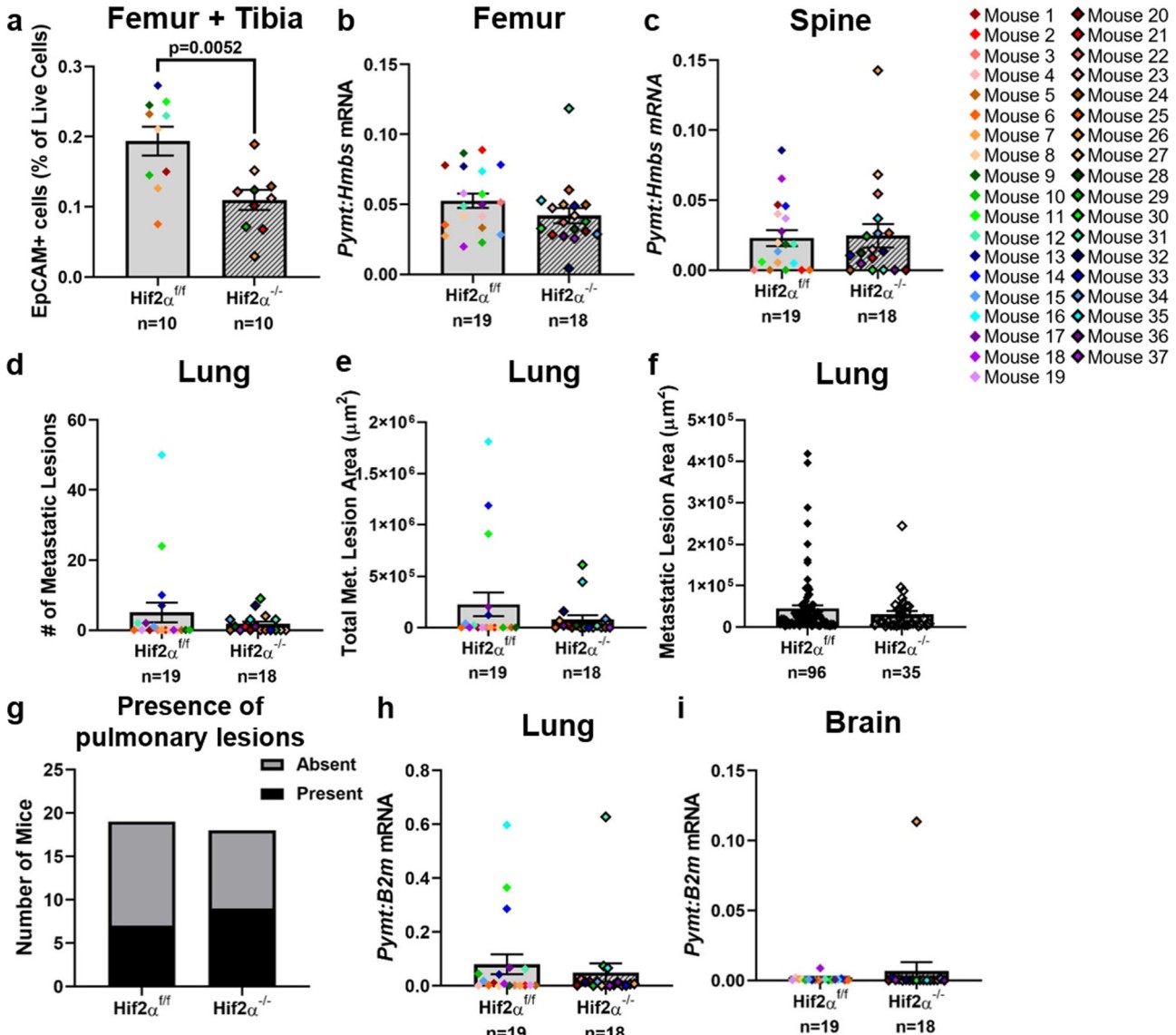

**Fig. 5 Deletion of *Hif2α* decreases tumor dissemination to the bone but not to the lung. a** The percentage of EpCAM$^+$ cells, out of the total number of live cells, detected by flow cytometric analysis of left hindlimb bone marrow. Two-tailed Mann–Whitney test. $n = 10$ Hif2α$^{f/f}$ PyMT$^+$ or Hif2α$^{-/-}$ PyMT$^+$ mice. **b**, **c** Quantitative PCR analysis of *Pymt* transcript compared to *Hmbs* from right femur or spinal midsection, respectively. Two-tailed Mann–Whitney test. **d**, **e** Metastatic lesion number or area detected by histological analysis of H&E-stained sections from the left lung. Two-tailed Mann–Whitney test. **f** Comparison of individual lesion areas detected from histological inspection of left lung sections. Two-tailed Mann–Whitney test. $n = 96$ individual Hif2α$^{f/f}$ metastatic lung lesions, $n = 35$ individual Hif2α$^{-/-}$ metastatic lung lesions. **g** Comparison of the proportion of mice from each group that had any detectable pulmonary lesions. Fisher's exact test. **h**, **i** Quantitative PCR analysis of *Pymt* transcript compared to *B2m* from the right lung or brain, respectively. Two-tailed Mann–Whitney test. Graphs represent mean per group and error bars represent s.e.m. For **b–e**, **g–i**, $n = 19$ Hif2α$^{f/f}$ PyMT$^+$ mice, $n = 18$ Hif2α$^{-/-}$ PyMT$^+$ mice.

be involved in the bone dissemination phenotypes we observed in Hif2α$^{-/-}$ PyMT$^+$ and Vhl$^{-/-}$ PyMT$^+$ mice. Our data suggest that reduced expression of PTHrP in the primary tumor may, in part, mediate the decreased bone dissemination in Hif2α$^{-/-}$ PyMT$^+$ mice, while increased C-X-C chemokine motif receptor 4 (CXCR4) expression may be involved in the regulation of bone dissemination in Vhl$^{-/-}$ PyMT$^+$ mice, although it is important to note that neither gene was significantly changed. PTHrP (gene name *PTHLH*) is a key driver of osteolysis in bone-disseminated breast cancer, but its role in the primary tumor is more nuanced[75]. Previous work using the MMTV-PyMT model demonstrated that PTHrP deletion in the mammary epithelium delays primary tumor initiation, inhibits tumor progression, and reduces metastasis to distal sites[51]. Loss of primary tumor PTHrP

expression in a MMTV-*neu* model of breast cancer, however, showed the opposite outcome, with PTHrP loss resulting in higher tumor incidence[76]. Clinical studies have also shown that patients with PTHrP-positive primary tumors have significantly improved survival and fewer metastases to distant sites, including the bone[77,78]. PTHrP is known to be expressed in a HIF2α, but not HIF1α, dependent manner[55]. In accordance with the modest decrease in *Pthlh* expression we observe in Hif2α$^{-/-}$ tumors, the trend toward increased *Pthlh* expression in Vhl$^{-/-}$ tumors may be due to Hif2α activation rather than Hif1α. CXCR4 is known to drive breast cancer metastasis to bone through engagement with its cognate ligand, CXCL12, which is highly expressed on mesenchymal stromal cells in the bone marrow[56,57]. Interestingly, although CXCR4 expression is known to be HIF1α dependent[39],

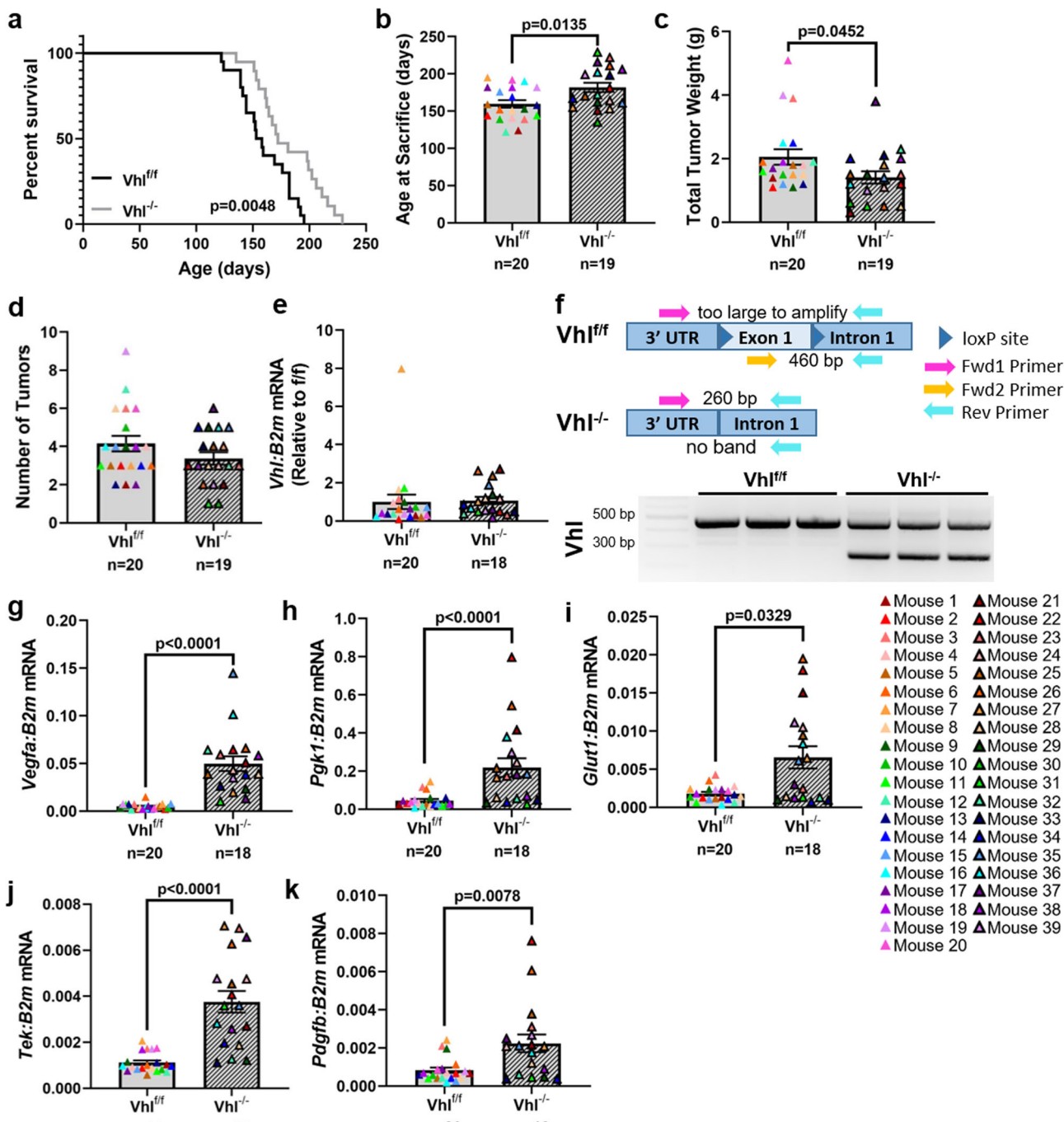

**Fig. 6 Deletion of *Vhl* slows primary tumor growth and decreases total tumor burden. a** Survival analysis of Vhl^f/f PyMT+ and Vhl^−/− PyMT+ mice where end point represents sacrifice due to tumor size reaching collection threshold. Log-rank test. **b**–**d** Comparison of the age at sacrifice, total burden at sacrifice, and number of tumors collected per mouse. Two-tailed Mann–Whitney test. **e** Quantitative PCR analysis of *Vhl* expression compared to *B2m* from whole-tumor homogenate RNA. Expression is normalized to the mean of the f/f control group. Two-tailed Mann–Whitney test. **f** Schematic of PCR-based validation of *Vhl* locus recombination using a three-primer PCR reaction. Genomic DNA was used as the input material. DNA electrophoresis gel represents the PCR products from the reactions depicted in the schematic diagram. The lower molecular weight of the band in the Vhl^−/− lanes indicate successful recombination. The residual band in the Vhl^−/− lanes are likely from non-recombined stromal cells present in the tumor. **g**–**k** Quantitative PCR analysis of HIF target genes (*Vegfa, Pgk1, Glut1, Tek, Pdgfb*) compared to *B2m* from whole-tumor homogenate RNA. Two-tailed Mann–Whitney test. Graphs represent mean per group and error bars represent s.e.m. n = 20 Vhl^f/f PyMT+ mice, n = 19 (or 18 in **e**–**k**) Vhl^−/− PyMT+ mice.

we did not observe a decrease in *Cxcr4* transcript in response to *Hif1α* deletion. There is also evidence that HIF2α is required for CXCR4 expression in renal cancer cell lines[79], but we did not observe a significant difference in *Cxcr4* expression in response to *Hif2α* deletion. This suggests that, in our model, deletion of either

of the HIFα factors on their own was not sufficient to decrease *Cxcr4*, but the combined stabilization of Hif1α and Hif2α may have been sufficient to drive *Cxcr4* expression. Alternatively, the modest increase in *Cxcr4* in response to *Vhl* deletion could be mediated by HIF-independent pathways. While VHL functions to

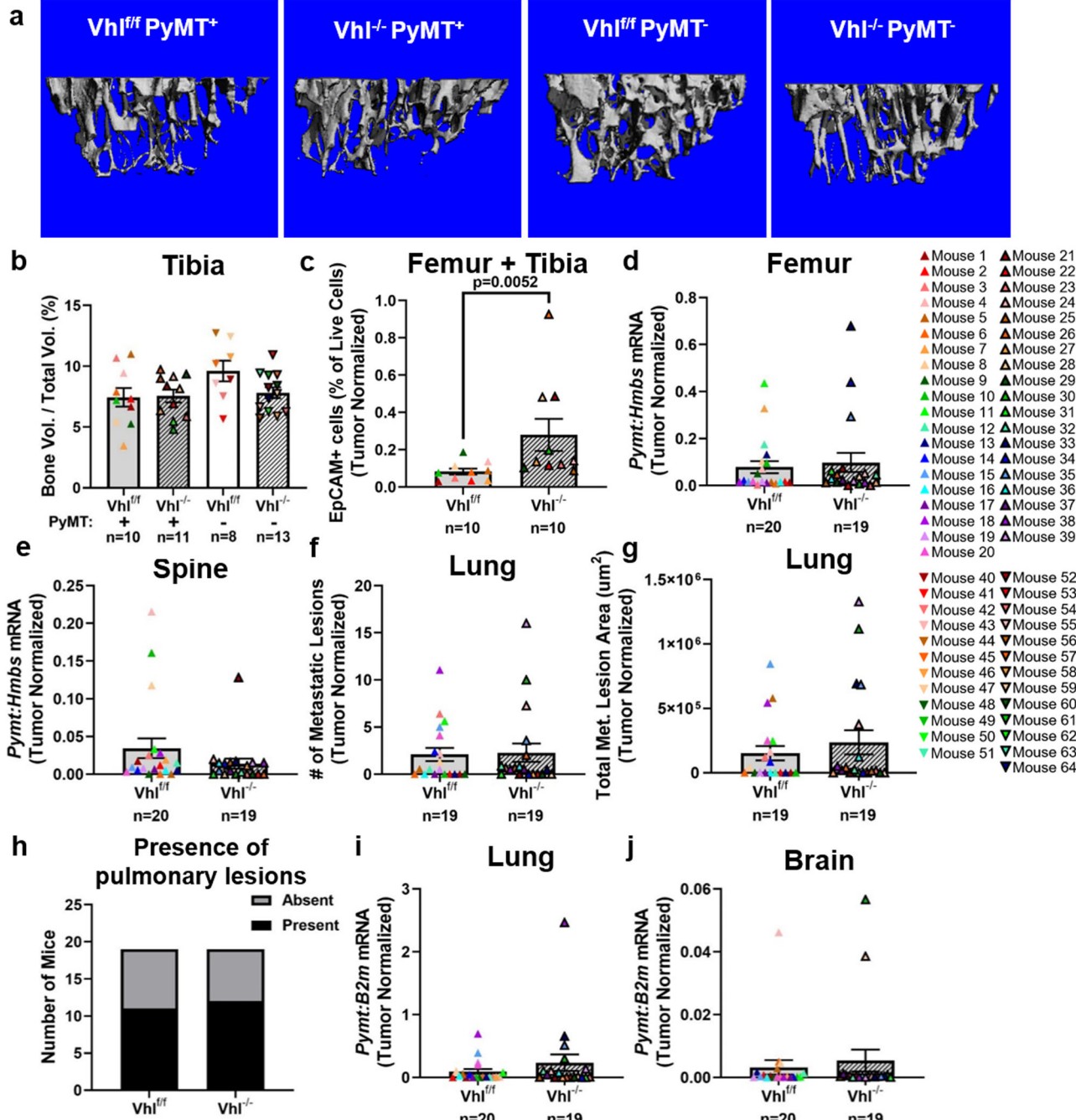

**Fig. 7 Deletion of *Vhl* increases tumor dissemination to the bone but not to the lung. a** Representative 3D renderings of microCT scans of the proximal metaphysis of the right tibia. **b** Quantification of bone volume from microCT analysis as a percentage of total volume. Two-tailed Mann–Whitney test against corresponding f/f control. $n = 10$ Vhl$^{f/f}$ PyMT$^+$ mouse tibiae, $n = 11$ Vhl$^{-/-}$ PyMT$^+$ mouse tibiae, $n = 8$ Vhl$^{f/f}$ PyMT$^-$ mouse tibiae, and $n = 13$ Vhl$^{-/-}$ PyMT$^-$ mouse tibiae. **c** The percentage of EpCAM$^+$ cells, out of the total number of live cells, detected by flow cytometric analysis of left hindlimb bone marrow. Values have been normalized to the total tumor burden at end point of each mouse. Two-tailed Mann–Whitney test. $n = 10$ Vhl$^{f/f}$ PyMT$^+$ mice, $n = 10$ Vhl$^{-/-}$ PyMT$^+$ mice. **d, e** Quantitative PCR analysis of *Pymt* transcript compared to *Hmbs* from right femur or spinal midsection, respectively. Values have been normalized to the total tumor burden at end point of each mouse. Two-tailed Mann–Whitney test. **f, g** Metastatic lesion number or area detected by histological analysis of H&E-stained sections from the left lung. Values have been normalized to the total tumor burden at end point of each mouse. Two-tailed Mann–Whitney test. **h** Comparison of the proportion of mice from each group that had any detectable pulmonary lesions. Fisher's exact test. **i, j** Quantitative PCR analysis of *Pymt* transcript compared to *B2m* from the right lung or brain, respectively. Values have been normalized to the total tumor burden at end point of each mouse. Two-tailed Mann–Whitney test. Graphs represent mean per group and error bars represent s.e.m. For **d**–**j**, $n = 20$ (or 19 in **f**–**h**) Vhl$^{f/f}$ PyMT$^+$ mice, $n = 19$ Vhl$^{-/-}$ PyMT$^+$ mice.

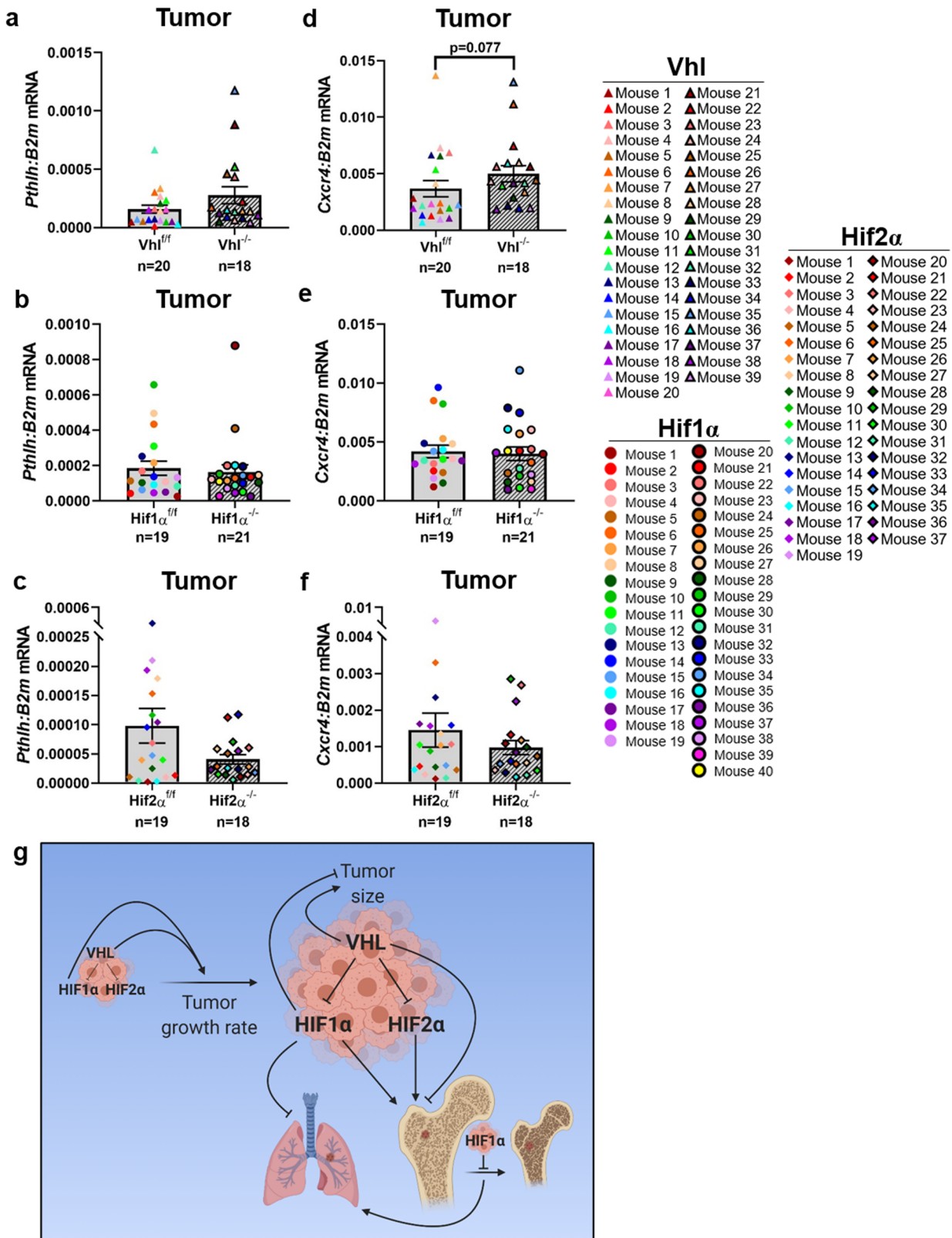

regulate HIF signaling activity by modulating degradation of HIF1α and HIF2α, VHL also possesses HIF-independent functions[80,81]. For example, VHL has been shown to degrade or inhibit the catalytic activity of certain kinases, including some PKC isoforms[82–84], AKT1, and AKT2[85]. These kinases are involved in signaling pathways that regulate metabolism, angiogenesis, proliferation, differentiation, cell survival, tumor initiation, and metastasis[86–88]. Thus, it is possible that phenotypes observed in the Vhl$^{-/-}$ PyMT$^+$ mice may be due, in part, to these HIF-independent pathways. To definitively address whether the phenotype and potential mechanisms observed in the Vhl$^{-/-}$ mice are specific for Hif1α or Hif2α, loss-of-function Vhl/Hif1α and Vhl/Hif2α double knockout mice will need to be investigated in future studies.

**Fig. 8 HIF signaling in the primary tumor drives dissemination to the bone but only modestly alters factors known to regulate bone metastasis.**
**a–c** Quantitative PCR analysis of *Pthlh* transcript compared to *B2m* in primary tumor with deletion of *Vhl*, *Hif1α*, or *Hif2α*, respectively. Two-tailed Mann–Whitney test. **d–f** Quantitative PCR analysis of *Cxcr4* transcript compared to *B2m* in primary tumor with deletion of *Vhl*, *Hif1α*, or *Hif2α*, respectively. Two-tailed Mann–Whitney test. **g** Summary of findings. VHL and HIF1α expression in the primary tumor drive tumor growth, and VHL expressing tumors are larger, while HIF1α-expressing tumors are smaller on average. HIF1α expression in the primary tumor also disrupts bone homeostasis, leading to decreased bone volume. This disruption of the bone inhibits tumor cell dissemination to bone, while likely promoting tumor cell dissemination and outgrowth in the lung. HIF2α drives bone dissemination but does not influence lung metastasis. As the negative regulator of HIF signaling, VHL inhibits bone metastasis but VHL expression does not alter metastatic tumor burden in the lung. Image created with BioRender.com. Graphs represent mean per group and error bars represent s.e.m. $n = 20$ Vhl$^{f/f}$ PyMT$^+$ mice, $n = 18$ Vhl$^{-/-}$ PyMT$^+$ mice, $n = 19$ Hif1α$^{f/f}$ PyMT$^+$ mice, $n = 21$ Hif1α$^{-/-}$ PyMT$^+$ mice, $n = 19$ Hif2α$^{f/f}$ PyMT$^+$ mice, $n = 18$ Hif2α$^{-/-}$ PyMT$^+$ mice.

Our findings present important considerations for the development of clinical HIF inhibitors. While targeted inhibition of HIF1α or HIF2α could be beneficial in preventing bone metastasis, HIF1α inhibition may promote the outgrowth of lung-disseminated breast cancer cells. Thus, the site-specific effects of tumor cell HIF signaling must be considered. Furthermore, our results suggest that patients with high levels of HIF1α or HIF2α expression in their primary tumor may be at an increased risk of bone metastasis development. Thus, these markers may have prognostic value and patients with high expression may benefit from bone-focused clinical follow-up over time.

## Methods

**Animals**. All experiments were performed following the relevant guidelines and regulations of the Animal Welfare Act and the Guide for the Care and Use of Laboratory Animals and were approved by the Institutional Animal Care and Use Committee at Vanderbilt University.

Mammary epithelial tumor cell-specific knock-out of *Hif1α* was achieved by breeding together three transgenic mouse strains. First, transgenic mice with loxP sites flanking both alleles of the Hif1α exon 2 (Jackson Laboratory Stock No. 007561, C57/B6 background)[89] were bred with transgenic mice expressing Cre recombinase downstream of MMTV-LTR (Jackson Laboratory Stock No. 003553, C57/B6 background)[90]. The progeny from this cross were then bred with transgenic mice expressing the PyMT oncoprotein under the MMTV-LTR (Jackson Laboratory Stock No. 022974, C57/B6-FVB mixed background)[91]. Virgin female mice with Hif1α wild-type mammary fat pad tumors (Hif1α$^{f/f}$ PyMT$^+$ = Hif1α$^{f/f}$, MMTV-Cre negative, MMTV-PyMT positive) or Hif1α-null mammary fat pad tumors (Hif1α$^{-/-}$ PyMT$^+$ = Hif1α$^{f/f}$, MMTV-Cre positive, MMTV-PyMT positive) were used in these studies. Non-tumor bearing (PyMT$^-$) controls of both Hif1α$^{f/f}$ and Hif1α$^{-/-}$ mice were also used.

Mammary epithelial tumor cell-specific knock-out of *Hif2α* was achieved using the same breeding strategy as above but using a transgenic mouse line with loxP sites flanking both alleles of the Hif2α exon 2 (Jackson Laboratory Stock No. 008407, C57/B6 background)[92]: Hif2α$^{f/f}$ PyMT$^+$ = Hif2α$^{f/f}$, MMTV-Cre negative, MMTV-PyMT positive; Hif2α$^{-/-}$ PyMT$^+$ = Hif1α$^{f/f}$, MMTV-Cre positive, MMTV-PyMT positive.

Similarly, mammary epithelial tumor cell-specific knock-out of *Vhl* was achieved by incorporating a transgenic mouse line with loxP sites flanking both alleles of the Vhl exon 1 (Jackson Laboratory Stock No. 012933, C57/B6 background)[93]: Vhl$^{f/f}$ PyMT$^+$ = Vhl$^{f/f}$, MMTV-Cre negative, MMTV-PyMT positive; Vhl$^{-/-}$ PyMT$^+$ = Vhl$^{f/f}$, MMTV-Cre positive, MMTV-PyMT positive.

At weaning, tail snips were collected from each mouse. Genomic DNA was extracted from the tail snip and genotyping was performed by PCR amplification of Hif1α (F: GGAGCTATCTCTCTAGACC, R: GCAGTTAAGAGCACTAGTTG)[94], Hif2α (F: CTTCTTCCATCATCTGGGATCTGGGAC, R: CAGGCAGTATGCC TGGCTAATTCCAGTT)[92], or Vhl (F: CTAGGCACCGAGCTTAGAGGTTGCG, R: CTGACTTCCACTGATGCTTGTCACAG)[94], as well as PyMT (F: GGAA GCAAGTACTTCACAAGGG, R: GGAAAGTCACTAGGAGCAGGG), and Cre (F: GCGGTCTGGCAGTAAAAACTATC, R: GTGAAACAGCATTGCTGTC ACTT). The PyMT genotyping reaction also included an internal positive control reaction (F: CAAATGTTGCTTGTCTGGTG, R: GTCAGTCGAGTGCACAG TTT). Primer sequences for PyMT and Cre genotyping were obtained from Jackson Laboratories. Once weaned, mice were palpated 2-3 times per week, and tumors were measured in three dimensions with digital calipers. Mice were collected when any tumor had grown to a size of 1 cm in diameter in any dimension, around 4–5 months of age. Littermate and cousin control mice were collected for each study. Forty-five minutes prior to collection, mice were inoculated with Hypoxyprobe (Hypoxyprobe-1 Omni Kit, Hypoxyprobe, Inc., catalog number HP3-1000Kit) via intraperitoneal injection at a dosage of 60 mg/kg body weight.

**Real-time PCR**. Tumor samples, right lung lobes, brains, and intact right femora were homogenized in 1 ml TRIzol (Life Technologies), spun down to clear the lysate, phenol–chloroform extracted, DNase digested (TURBO DNA-free Kit, Life Technologies), and cDNA synthesized (1 μg RNA, iScript cDNA Synthesis Kit, Bio-Rad) per the manufacturer's instructions. Real-time PCR was performed using iTaq Universal SYBR Green Supermix (Bio-Rad) on a QuantStudio 5 (Thermo Fisher) with the following conditions: 2 min at 50 °C, 10 min at 95 °C, (15 s at 95 °C, 1 min at 60 °C) × 40 cycles followed by dissociation curve (15 s at 95 °C, 1 min at 60 °C, 15 s at 95 °C). For each biological replicate, three technical replicates were performed for each gene analyzed. Non-template controls were included as a negative control for each gene analyzed. Analysis was performed by normalizing the expression of the target gene to the average *Hmbs* or *B2m* expression within the same sample to determine $\Delta$Ct. The $\Delta$Ct was transformed ($2^{-\Delta Ct}$) and the average of the three technical replicates was calculated. The average $2^{-\Delta Ct}$ for each mouse is presented as target gene "(*Pymt, Hif1a, Hif2a*, etc.): (*B2m* or *Hmbs*) mRNA" or "Relative mRNA abundance" in the figures. Mouse primers for PyMT[95] (*Pymt*—F: CTGCTACTGCACCCAGACAA, R: GCAGGTAAGAGGCATTCTGC) and hydroxymethylbilane synthase[96] (*Hmbs*—F: TCATGTCCGGTAACGGCG, R: CACTCGAATCACCCTCATCTTTG) were previously published. Primer sequences for parathyroid hormone-related protein (*Pthlh*—F: ACATTGCTATGG-GAGCCAC, R: TAGGAATCAGCGCCTCTAAC) were kindly provided by Dr. Natalie Sims and Dr. T. John Martin at St. Vincent's Institute of Medical Research. Primers for beta-2-microglobulin, VEGFA, Hif1α (exon 2), Hif2α (exon 2), and Vhl (exon 1) were designed using PrimerBlast (NCBI) against the mouse genome (*Mus musculus*) and validated by dissociation: *B2m* (F: TTCACCCCCACTGAGACTG AT, R: GTCTTGGGCTCGGCCATA), *Vegfa* (F: GGAGATCCTTCGAGGAGCAC TT, R: GGCGATTTAGCAGCAGATATAAGAA), *Hif1α* (F: TCGGCGAAGCAA AGAGTCTG, R: GCTCACATTGTGGGGAAGTG), *Hif2α* (F: TGAGGAAGGAG AAATCCCGTG, R: GGGCAACTCATGAGCCAACT), *Vhl* (F: GACCCGTTCCA ATAATGCCC, R: CGTCGAAGTTGAGCCACAAA). Additional primer sequences were obtained from the Massachusetts General Hospital PrimerBank and specificity was confirmed by nucleotide BLAST (NCBI) and validated by dissociation: *Cdh1* (F: CAGGTCTCCTCATGGCTTTGC, R: CTTCCGAAAAGAA GGCTGTCC), *Cdh2* (F: AGCGCAGTCTTACCGAAGG, R: TCGCTGCTTTCAT ACTGAACTTT), *Egfr* (F: ATGAAAACACCTATGCCTTAGCC, R: TAAGTT CCGCATGGGCAGTTC), *Notch1* (F: GATGGCCTCAATGGGTACAAG, R: TCGTTGTTGTTGATGTCACAGT), *Snai1* (F: CACACGCTGCCTTGTGTCT, R: GGTCAGCAAAAGCACGGTT), *Snai2* (F: CATCCTTGGGGCGTGTAAGTC, R: GCCCAGGAAACGTAGAATAGGTC), *Snai3* (F: GGTCCCCAACTACGGG AAAC, R: CTGTAGGGGGTCACTGGGATT), *Twist1* (F: GGACAAGCTG AGCAAGATTCA, R: CGGAGAAGGCGTAGCTGAG), *Zeb1* (F: ACTGCAAG AAACGGTTTTCCC, R: GGCGAGGAACACTGAGATGT), *Zeb2* (F: ATTGCAC ATCAGACTTTGAGGAA, R: ATAATGGCCGTGTCGCTTCG), *Atg7* (F: TCTGGGAAGCCATAAAGTCAGG, R: GCGAAGGTCAGGAGCAGAA), *Becn1* (F: ATGGAGGGGTCTAAGGCGTC, R: TCCTCTCCTGAGTTAGCCTCT), *Map1lc3a* (F: GACCGCTGTAAGGAGGTGC, R: CTTGACCAACTCGCTCA TGTTA), *Map1lc3b* (F: TTATAGAGCGATACAAGGGGGAG, R: CGCCGTCT GATTATCTTGATGAG), *Sqstm1* (F: GAACTCGCTATAAGTGCAGTGT, R: AGAGAAGCTATCAGAGAGGTGG), *Ulk1* (F: AAGTTCGAGTTCTCTCGC AAG, R: CGATGTTTTCGTGCTTTAGTTCC), *Pgk1* (F: TGGTGGGTGTGA ATCTGCC, R: ACTTTAGCGCCTCCCAAGATA), *Glut1* (F: CAGTTCGGCTAT AACACTGGTG, R: GCCCCCGACAGAGAAGATG), *Tek* (F: CTGGAGGTT ACTCAAGATGTGAC, R: TCCGTATCCTTATAGCCTGTCC), *Pdgfb* (F: CATCCGCTCCTTTGATGATCTT, R: GTGCTCGGGTCATGTTCAAGT), *Cxcr4* (F: GAGGCCAAGGAAACTGCTG, R: GCGGTCACAGATGTACCCGTC). The expression of a panel of metastasis-associated genes was quantified using the tumor metastasis (SAB Target List, M384) qPCR array plate (Bio-Rad). Each gene in the array was run in duplicate. Three representative Hif1α$^{f/f}$ and Hif1α$^{-/-}$ tumor homogenate RNA samples were selected based on their *Hif1α* expression. Analysis was performed by normalizing the expression of the target gene to the average *B2m* expression within the same sample to determine $\Delta$Ct. The $\Delta$Ct was transformed ($2^{-\Delta Ct}$) and the average of the two technical replicates was calculated. $\Delta$Ct values for each Hif1α$^{-/-}$ sample was then normalized against the average of the three Hif1α$^{f/f}$ samples to determine enrichment.

**Polymerase chain reaction**. Recombination and loss of the *Hif1α* exon 2 was confirmed by PCR amplification using a three-primer system split between two separate reactions. The forward primer (Fwd: CAGTGCACAGAGCCTCCTC) binds to *Hif1α* exon 1, the first reverse primer (Rev1: GCTCACATTGTGGG GAAGTG) binds to exon 2, and the second reverse primer (Rev2: ATGTAAAC CATGTCGCCGTC) binds to exon 3. The first reaction (Fwd and Rev1 primers) will not amplify a band if recombination has occurred, while the second reaction (Fwd and Rev2) will amplify a lower molecular weight band in the case of recombination. PCR reactions were set up using the HotStarTaq system (Qiagen) using 200 ng of cDNA generated from RNA extracted from tumor homogenates as the template. PCR reactions were carried out in a T100 Thermal Cycler (Bio-Rad) with the following conditions: 15 min at 95 °C, (1 min at 94 °C, 1.5 min at 59 °C, 1 min at 72 °C) × 38 cycles, followed by 10 min at 72 °C. The *B2m* locus was also amplified as a loading control, using the same primers as used for real-time PCR above, using the following conditions: 15 min at 95 °C, (1 min at 94 °C, 1.5 min at 53 °C, 1 min at 72 °C) × 35 cycles, followed by 10 min at 72 °C.

Recombination of *Vhl* exon 1 was confirmed by PCR amplification using a one-reaction three-primer system[97]. The first forward primer (Fwd1: CTGGTACCCAC GAAACTGTC) binds to the 3' UTR upstream of *Vhl* exon 1, the second forward primer (Fwd2: CTAGGCACCGAGCTTAGAGGTTTGCG) binds to *Vhl* exon 1, and the reverse primer (Rev: CTGACTTCCACTGATGCTTGTCACAG) binds to a region in the first intron. If no recombination has occurred, only one fragment will amplify (Fwd2 and Rev primer pair) since the amplification fragment that would form from the Fwd1 and Rev primer pair is too large to amplify. If recombination has occurred, the binding site for the Fwd2 primer is no longer present, and a lower molecular weight band will form (Fwd1 and Rev primers). PCR reactions were set up using the HotStarTaq system (Qiagen) using 200 ng of cDNA generated from RNA extracted from tumor homogenates as the template. PCR reactions were carried out in a T100 Thermal Cycler (Bio-Rad) with the following conditions: 15 min at 95 °C (1 min at 94 °C, 1.5 min at 52 °C, 1 min at 72 °C) × 35 cycles, followed by 10 min at 72 °C.

**Histology**. All tissues collected were fixed in 10% formalin for 24 h. Hind limb bones were then decalcified in EDTA (20% pH 7.4) solution for 72 h prior to embedding. Tissues analyzed via histology and immunohistochemistry were embedded in paraffin and 5 µm-thick sections were prepared for staining. Hematoxylin and eosin (H&E) staining was performed[98] and H&E-stained slides of left lung, liver, and tibia sections were analyzed for the presence of metastatic mammary tumors under blinded conditions by a board-certified veterinary anatomic pathologist. All tumors were imaged on an Olympus BX43 microscope equipped with a SPOT Flex camera (Diagnostic Instruments Inc., Sterling Heights, MI). Tumor areas were measured in the SPOT software (Diagnostic Instruments Inc.), and total tumor area per animal was calculated.

**Pimonidazole immunohistochemistry**. Sections were deparaffinized and rehydrated by soaking the slides in Xylene Substitute (Thermo Fisher), 100, 95, 90, and then 70% ethanol, and finally deionized water. Peroxidase activity was then quenched by incubating the slides in 3% hydrogen peroxide for 15 min. After slowly displacing the hydrogen peroxide solution with deionized water, slides were rinsed three times with phosphate-buffered saline (PBS). The deparaffinized sections were blocked in serum-free protein blocking agent (Dako) for 5 min and incubated with primary antibody (Hypoxyprobe-1 Omni Kit, Hypoxyprobe, Inc., catalog number HP3-1000Kit, 1:100) in Dako protein block overnight at 4 °C. Negative control sections were incubated with Dako alone. The sections were then washed three times with PBS and incubated with biotinylated secondary antibody (goat-anti-rabbit IgG, Vector, catalog number BA-1000, 1:200) in Dako protein block for 30 min at 37 °C, followed by streptavidin-conjugated horseradish peroxidase (HRP; Millipore, OR03L, 1:200) in PBS for 30 min at 37 °C, and developed using the DAB ImmPACT Kit (Vector, catalog number SK-4105) for approximately 1 min. Chromogen development was then quenched by rinsing twice in deionized water. Sections were counterstained with hematoxylin and dehydrated by soaking them in 70, 90, 95, and 100% ethanol, followed by Xylene Substitute. The coverslips were mounted using Permount (Fisher Scientific). All images were collected on an Olympus BX41 Microscope equipped with an Olympus DP71 camera using the ×4, ×20, and ×40 objectives. Pimonidazole-positive tumor area was quantified based on visual inspection of multiple micrographs with ×4 objective based on a scale of 0–3: 0 = no nuclear staining, 1 = <10% positive nuclear staining, 2 = 10–50% positive nuclear staining, 3 = ≥50% positive nuclear staining. Each field was assigned a score and an average score for the tumor was calculated based on all fields. The number of fields inspected varied from 1 to 4, depending on the size of the tumor. Pimonidazole-stained area of lung lesions was quantified using the ImageJ software with manual tumor area contouring and automatic color thresholding.

**Ki-67 immunohistochemistry**. Staining was performed by the Vanderbilt University Medical Center Translational Pathology Shared Resource (Nashville, TN) as follows: Slides were placed on the Leica Bond Max IHC stainer. All steps besides dehydration, clearing, and coverslipping are performed on the Bond Max. Slides are deparaffinized. Heat-induced antigen retrieval was performed on the Bond Max

using their Epitope Retrieval 2 solution for 20 min. Slides were placed in a Protein Block (Ref# x0909, DAKO, Carpinteria, CA) for 10 min. The sections were incubated with anti-Ki-67 (Catalog #12202 S, Cell Signaling Technology, Danvers, MA) diluted 1:250 for 1 h. The Bond Refine Polymer detection system was used for visualization. Slides were the dehydrated, cleared, and coverslipped. All images were collected on an Olympus BX41 Microscope equipped with an Olympus DP71 camera using the ×4, ×20, and ×40 objectives. Ki-67-stained area of lung lesions was quantified using the ImageJ software with manual tumor area contouring and automatic color thresholding.

**CD4/CD8 immunofluorescent imaging**. Slides were stained using the Opal 7-Color Manual IHC Kit (PerkinElmer). In brief, slides were deparaffinized and rehydrated in a series of xylene and ethanol washes, microwaved in Rodent Decloaker (Biocare Medical), and then cooled. Slides were washed with water and then TBST (TBS + 0.05% Tween-20) followed by blocking with BLOXALL Blocking Solution (Vector Laboratories). The following antibodies were used: CD4 (Invitrogen, clone 4SM95; 1:100), CD8 (Invitrogen, clone 4SM16; 1:100), ImmPRESS HRP Goat anti-rat IgG (Vector Laboratories; 1:4). Staining with primary and secondary antibodies and OPAL fluorophores were performed per the manufacturer's instructions. All images were collected on a Zeiss 880/Airyscan Microscope using the ×20 and ×40 objectives. For quantitation, the ×20 images were used to manually count the number of CD4+ and CD8+ cells and calculate nuclear area using color thresholding in ImageJ.

**Microcomputed tomography**. Ex vivo microCT was performed on the proximal tibia using the Scanco µCT 50. Scans were initiated from the proximal end of the metaphyseal growth plate and progressed 250 slices distal. Tibiae were scanned at 7 µm voxel resolution, 55-kV voltage, and 200 µA current. Scans were reconstructed and analyzed using the Scanco Medical Imaging Software to determine the bone volume/total volume, trabecular number, thickness, and separation. The most distal slice of the growth plate was used as a reference slice and analysis was set to begin 20 slices distal from this point. A 150 slice region of interest was selected for analysis. An automated contouring procedure was applied to separate the trabecular bone from the cortical bone and visually verified for each sample.

**Flow cytometry**. The epiphyses of the femur and tibia from one hindlimb were cut and the bones were flushed using centrifugation to obtain the bone marrow. The bone marrow was filtered through a 40 µm cell strainer to separate the cells from bone debris. Cells were suspended in red blood cell lysis buffer for 5 min on ice, spun down, and washed twice with PBS. Bone marrow (2 × 10⁶ cells) was stained in 100 µl of 1% bovine serum albumin in PBS with 100 ng EpCAM antibody (BD Pharmingen, catalog number 563478) for 1 h at 4 °C in the dark. Cells were washed with PBS and resuspended in PBS and 0.5 ng 4,6-diamidino-2-phenylindole (DAPI) for 15 min on ice. Flow cytometric experiments were analyzed in the VUMC Flow Cytometry Shared Resource using the 3-laser or 5-laser BD LSRII. Datasets were analyzed using the FlowJo software (FlowJo, LLC). Cells were gated based on forward scatter and side scatter to identify single cells and then live cells (DAPI−) were gated based on EpCAM positivity using non-tumor-bearing bone marrow as a negative control. Mammary fat pad tumors were mechanically digested and filtered through a 40 µm cell strainer to obtain a single-cell suspension. These tumor cells were stained as described above and used as a positive control.

**TCGA patient data analysis**. The Cancer Genome Atlas cBioPortal[99,100] was accessed on 16 November 2020, 17 November 2020, and 15 April 2021 to determine whether the hypoxia gene signature[45] from *n* = 1100 patients with Breast Invasive Carcinoma (Firehose Legacy dataset; http://www.cbioportal.org/study/summary?id=brca_tcga) correlated with bone[46] and lung metastasis signatures[47]. Genes from the hypoxia, bone metastasis, and lung metastasis gene signatures were entered into the query gene field for cBioPortal separately. The mRNA expression (RNA Seq V2 RSEM) for each gene across all 1100 patients was downloaded and saved in Excel format. For each gene in the three gene signatures, the gene expression provided by RNAseq was normalized to the median expression of that gene across all patients[101]. We next averaged the expression score for all of the genes in that signature for each individual patient. A log conversion was applied to the normalized, averaged gene signature expression scores. These signature scores were plotted against each other as a correlation graph in the Prism software (Graphpad) with a linear regression curve.

**Statistics and reproducibility**. To calculate sample size, we previously consulted with a biostatistician within the Department of Radiation Oncology at Stanford University who performed power calculations. The power calculations were based on previously published in vivo studies[33]. For the transgenic mouse lines, a sample size of *n* = 10 mice/group will provide 85% power in a two-sided *t* test with an alpha level 0.05 with a variation of 8.5 and an expected difference of 12. Thus, we sought to collect a minimum of *n* = 10 mice/group to ensure adequate statistical analysis at end point for these experiments. For all studies, *n* per group is as indicated in the figure or figure legend and the scatter dot plots indicate the mean of each group and the standard error of the mean. All graphs and statistical analyses were generated using the Prism software (Graphpad). All in vitro and in vivo assays were analyzed for statistical significance using two-tailed

Mann–Whitney test, log-rank test, or Fisher's exact test, as denoted in figure legends. Pearson and Spearman correlation coefficients were calculated for gene signature correlation analyses. $P < 0.05$ was statistically significant.

**Reporting summary**. Further information on research design is available in the Nature Research Reporting Summary linked to this article.

## Data availability

Raw data underlying figures can be found in Supplementary Data 1. Unedited DNA gel images are available in the Supplementary Information (Supplementary Figs. 7–12). Any other data that support the findings of this study are available from the corresponding author, R.W.J. upon reasonable request.

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

## Acknowledgements

The authors wish to thank Mr. Joshua Johnson for bone histological processing, sectioning, and H&E staining. V.M.T. and R.W.J. are supported by NIH award R00CA194198 (to R.W.J.), and L.A.V., M.E.C., and R.W.J. are supported by DoD Breakthrough Award W81XWH-18-1-0029 (to R.W.J.). M.R. is supported by NIH award R00CA201304. This project was supported in part by scholarship funds from the NIH award P30CA068485 Vanderbilt-Ingram Cancer Center Support Grant, which also supports the Translational Pathology Shared Resource. Flow Cytometry experiments were performed in the VMC Flow Cytometry Shared Resource, which is supported by P30CA68485 and the Vanderbilt Digestive Disease Research Center (DK058404). The results shown here are in part based on data generated by the TCGA Research Network: https://www.cancer.gov/tcga.

## Author contributions

R.W.J. and V.M.T conceived the original idea. M.R. provided support and critical feedback to the project and manuscript. R.W.J. supervised the project and provided funding. V.M.T. designed and performed the majority of experiments and data analysis and wrote the manuscript. L.A.V. maintained animal colonies and performed the majority of the animal collections. K.P.S. and C.D.O. performed experiments and assisted V.M.T. L.H. and C.P. performed the lung and liver histology analysis. M.E.C. performed the immunofluorescence experiments and analysis.

## Competing interests

The authors declare no competing interests.

**Additional information**

