## [Transparent Peer Review File · Communications Biology]

Reviewers' comments:

Reviewer #1 (Remarks to the Author):

The manuscript by Todd et al. investigates the role of hypoxia inducible factors HIF-1 and HIF-2 in promoting breast cancer metastasis to the bone and lung using the PyMT-MMTV spontaneous tumor model. This manuscript uses multiple genetic mouse models to investigate the effects of HIF-1, HIF-2 or VHL deletion on the development of breast cancer bone and lung metastases. The major claims of the paper are that breast cancer cell HIF-1 or HIF-2 promote metastasis to the bone; whereas HIF-1 inhibits and HIF-2 does not contribute to metastasis to the lung.

1) The findings that breast cancer cell HIF-1 and HIF-2 promote metastasis to the bone are consistent to previous work. Previous studies have shown that the hypoxic gene signature is associated with bone metastasis in breast cancer patients (1, 2). In the human breast cancer cells, HIF-1 is necessary and sufficient for MDA-MB-231 cells to metastasize to the bone (3-5). As the authors note, the PyMT-MMTV model used in this study is not the most robust model to investigate breast to bone metastases. Individual cells identified by FACS sorting for EpCAM+ cells in the bone marrow revealed a modest, but significant decrease in tumor cells from the HIF-1 or HIF-2 deficient mice. However, specific real time PCR analysis for Pymt did not reveal differences between control, HIF-1 or HIF-2 deficient tumor burden.

An interesting observation in this study is the decrease in trabecular bone within the PyMT HIF-1 deficient mice. The data presented suggest that this may be due to a secreted factor from the primary tumor, but independent of LOX expression. The authors speculate that this may be due to altered PTHrP expression, but this is not investigated in this manuscript.

2) The findings that HIF-1 inhibits and HIF-2 does not contribute to breast to lung metastases is contradictory to a large body of work that supports a role for HIF signaling in promoting breast to lung metastases. Breast cancer patients at high risk of developing lung metastasis can be identified using a hypoxia response signature set of 45 genes (5). Preclinical studies using human breast cancer cells (MDA-MB-231) as well as the genetic mouse PyMT-MMTV model used in this study demonstrated that HIF-1 and HIF-2 drive the metastatic potential of breast cancer cells to the lung in vivo (5-8). In addition, multiple mechanisms have been identified by which HIF promote breast to lung metastases including the enhanced secretion of proteins including lysyl oxidase (LOX) and LOX like proteins (LOXL2 and LOXL4) to prime the lung microenvironment for colonization (9, 10). HIF signaling also directly facilitates tumor extravasation to the lung through the upregulation of factors that promote tumor-endothelial cell adhesions or decrease endothelial cell-cell adhesion (7).

One possible explanation for the observation that HIF-1 inactivation increases lung metastasis in this study may be that the HIF-1 knockout mice were sacrificed on average 20 days (in some cases >40 days) later than the control animals. This additional time may have allowed for additional time for the breast cancer cells to seed and/or grow in the lung. The authors noted that they normalized the metastatic tumor burden in the lung to endpoint primary tumor burden. However, normalization to time at sacrifice is necessary as metastatic potential is not always correlated with primary tumor size.

References:

1. Woelfle U, Cloos J, Sauter G, Riethdorf L, Janicke F, van Diest P, Brakenhoff R, Pantel K. Molecular signature associated with bone marrow micrometastasis in human breast cancer. *Cancer research*. 2003;63(18):5679-84. Epub 2003/10/03. PubMed PMID: 14522883.
2. Cox TR, Rumney RM, Schoof EM, Perryman L, Hoye AM, Agrawal A, Bird D, Latif NA, Forrest H, Evans HR, Huggins ID, Lang G, Linding R, Gartland A, Erler JT. The hypoxic cancer secretome induces pre-metastatic bone lesions through lysyl oxidase. *Nature*. 2015;522(7554):106-10. Epub 2015/05/29. doi: 10.1038/nature14492. PubMed PMID: 26017313; PMCID: 4961239.
3. Hiraga T, Kizaka-Kondoh S, Hirota K, Hiraoka M, Yoneda T. Hypoxia and hypoxia-inducible factor-1 expression enhance osteolytic bone metastases of breast cancer. *Cancer research*. 2007;67(9):4157-63. Epub 2007/05/08. doi: 10.1158/0008-5472.CAN-06-2355. PubMed PMID: 17483326.

4. Dunn LK, Mohammad KS, Fournier PG, McKenna CR, Davis HW, Niewolna M, Peng XH, Chirgwin JM, Guise TA. Hypoxia and TGF-beta drive breast cancer bone metastases through parallel signaling pathways in tumor cells and the bone microenvironment. *PloS one*. 2009;4(9):e6896. Epub 2009/09/04. doi: 10.1371/journal.pone.0006896. PubMed PMID: 19727403; PMCID: 2731927.
5. Lu X, Yan CH, Yuan M, Wei Y, Hu G, Kang Y. In vivo dynamics and distinct functions of hypoxia in primary tumor growth and organotropic metastasis of breast cancer. *Cancer research*. 2010;70(10):3905-14. Epub 2010/05/06. doi: 10.1158/0008-5472.CAN-09-3739. PubMed PMID: 20442288; PMCID: 2872139.
6. Liao D, Corle C, Seagroves TN, Johnson RS. Hypoxia-inducible factor-1alpha is a key regulator of metastasis in a transgenic model of cancer initiation and progression. *Cancer research*. 2007;67(2):563-72. Epub 2007/01/20. doi: 10.1158/0008-5472.CAN-06-2701. PubMed PMID: 17234764.
7. Zhang H, Wong CC, Wei H, Gilkes DM, Korangath P, Chaturvedi P, Schito L, Chen J, Krishnamachary B, Winnard PT, Jr., Raman V, Zhen L, Mitzner WA, Sukumar S, Semenza GL. HIF-1-dependent expression of angiopoietin-like 4 and L1CAM mediates vascular metastasis of hypoxic breast cancer cells to the lungs. *Oncogene*. 2012;31(14):1757-70. Epub 2011/08/24. doi: 10.1038/onc.2011.365. PubMed PMID: 21860410; PMCID: 3223555.
8. Goto Y, Zeng L, Yeom CJ, Zhu Y, Morinibu A, Shinomiya K, Kobayashi M, Hirota K, Itasaka S, Yoshimura M, Tanimoto K, Torii M, Sowa T, Menju T, Sonobe M, Takeya H, Toi M, Date H, Hammond EM, Hiraoka M, Harada H. UCHL1 provides diagnostic and antimetastatic strategies due to its deubiquitinating effect on HIF-1alpha. *Nature communications*. 2015;6:6153. Epub 2015/01/24. doi: 10.1038/ncomms7153. PubMed PMID: 25615526; PMCID: 4317501.
9. Erler JT, Bennewith KL, Nicolau M, Dornhofer N, Kong C, Le QT, Chi JT, Jeffrey SS, Giaccia AJ. Lysyl oxidase is essential for hypoxia-induced metastasis. *Nature*. 2006;440(7088):1222-6. Epub 2006/04/28. doi: 10.1038/nature04695. PubMed PMID: 16642001.
10. Wong CC, Gilkes DM, Zhang H, Chen J, Wei H, Chaturvedi P, Fraley SI, Wong CM, Khoo US, Ng IO, Wirtz D, Semenza GL. Hypoxia-inducible factor 1 is a master regulator of breast cancer metastatic niche formation. *Proceedings of the National Academy of Sciences of the United States of America*. 2011;108(39):16369-74. Epub 2011/09/14. doi: 10.1073/pnas.1113483108. PubMed PMID: 21911388; PMCID: 3182724.

Reviewer #2 (Remarks to the Author):

In the paper "Hypoxia inducible factor signaling in breast tumors controls spontaneous tumor dissemination in a site-specific manner", Todd and colleagues used a PyMT-driven spontaneous mouse mammary carcinoma model with mammary specific Hif1a, Hif2a, or Vhl knockout to investigate whether HIF signaling controls site-specific breast tumor metastasis. Although this work provides some interesting observations, the data presented is not solid enough to back up the conclusion.

1. It is will established that Vhl degradation results in Hif1 and Hif2 protein stabilization upon hypoxia. It is surprising to see no examination of Hif1a and Hif2a protein level in the primary tumors, bone and lung metastasis using Western blot or IHC. This data is particular important when performing experiments using the Vhl knockout mice.
2. Follow up from the previous point, Vhl not only regulates Hifs but also numerous factors in the cancer cells (as the authors described in the discussion). Since the authors used Vhl knockout as a Hif1/Hif2 gain-of-function model, it is necessary to use Vhl/Hif1a or Vhl/Hif2a double-knockout mice to demonstrate that the effects observed in Vhl knockout is Hif1 or Hif2 specific.
3. Most of the results in this manuscript showed no differences between the control and knockout. It

could be due to the sample collection time. Mice were sacrificed when tumor size reached 1 cm in diameter in the mammary gland. May be a longer time (when the tumor reach 2 cm) will show a more dramatic effect? Or a shorter time point can capture the difference between the knockout and the control? The authors are encouraged to perform a time course experiment to detail characterize the kinetic difference in bone and lung metastasis in the knockout animal compared to the control.

4. In figure 4, expression of EMT-related genes in primary tumors were examined and no difference was found between the control and Hif1a knockout group. This is not surprising as HIF2 and other hypoxia induced signaling pathways are known to upregulate EMT. What's confusing is why the authors used hypoxia gene signature identified in kidney cancer (Li et al) to correlate with bone and lung metastasis in breast cancer? Please check this reference for 42-gene hypoxia signature in breast cancer (PMID: 30037853).

5. On page 9, line 168: "the increased lung tumor burden is due to lung microenvironment-specific factors promoting outgrowth of disseminated tumor cells." If the Hif knockout is mammary tissue specific, what will be the lung specific factors between the control and knockout mice that promote the metastatic tumor growth?

Minor:

1. In the discussion section, there is a typo on line 259: It should be "Hif1a" not "Hi12a".

2. The papers cited in the introduction are not up to date. There are many recent research and review discussing Hifs and cancer metastasis. The authors should cite more recent papers.

Response to Reviewers:

We thank the reviewers for their thoughtful and constructive feedback. Please see below for our point-by-point responses.

Reviewer #1 (Remarks to the Author):

The manuscript by Todd et al. investigates the role of hypoxia inducible factors HIF-1 and HIF-2 in promoting breast cancer metastasis to the bone and lung using the PyMT-MMTV spontaneous tumor model. This manuscript uses multiple genetic mouse models to investigate the effects of HIF-1, HIF-2 or VHL deletion on the development of breast cancer bone and lung metastases. The major claims of the paper are that breast cancer cell HIF-1 or HIF-2 promote metastasis to the bone; whereas HIF-1 inhibits and HIF-2 does not contribute to metastasis to the lung.

1) The findings that breast cancer cell HIF-1 and HIF-2 promote metastasis to the bone are consistent to previous work. Previous studies have shown that the hypoxic gene signature is associated with bone metastasis in breast cancer patients (1, 2). In the human breast cancer cells, HIF-1 is necessary and sufficient for MDA-MB-231 cells to metastasize to the bone (3-5). As the authors note, the PyMT-MMTV model used in this study is not the most robust model to investigate breast to bone metastases. Individual cells identified by FACS sorting for EpCAM+ cells in the bone marrow revealed a modest, but significant decrease in tumor cells from the HIF-1 or HIF-2 deficient mice. However, specific real time PCR analysis for Pymt did not reveal differences between control, HIF-1 or HIF-2 deficient tumor burden. An interesting observation in this study is the decrease in trabecular bone within the PyMT HIF-1 deficient mice. The data presented suggest that this may be due to a secreted factor from the primary tumor, but independent of LOX expression. The authors speculate that this may be due to altered PTHrP expression, but this is not investigated in this manuscript.

Response: Thank you for these comments. We have re-emphasized in the discussion that the decrease in bone metastasis observed in the $Hif1\alpha^{-/-}$ and $Hif2\alpha^{-/-}$ mice is consistent with prior studies (page 14). We agree that the decrease in trabecular bone is interesting and propose that this may be due to a secreted factor other than LOX, but that the molecular mechanism remains unknown at this time. To clarify, we proposed that reduced PTHrP may mediate the reduction in bone dissemination in the $Hif2\alpha^{-/-}$ mice, not the reduced trabecular bone in the $Hif1\alpha^{-/-}$ mice (please see pages 18 and 19 in the discussion). We have attempted immunostaining for PTHrP in the primary tumors of the $Hif1\alpha^{-/-}$ and $Vhl^{-/-}$ mice since we had these sections available but were unable to stain for PTHrP above background (data not shown). The reagents to detect PTHrP are notoriously unreliable, given that PTHrP has multiple promoters and cleavage sites, and we therefore focused on the mRNA expression of the molecule, which was included in Figure 8A-C.

2) The findings that HIF-1 inhibits and HIF-2 does not contribute to breast to lung metastases is contradictory to a large body of work that supports a role for HIF signaling in promoting breast to lung metastases. Breast cancer patients at high risk of developing lung metastasis can be identified using a hypoxia response signature set of 45 genes (5). Preclinical studies using human breast cancer cells (MDA-MB-231) as well as the genetic mouse PyMT-MMTV model used in this study demonstrated that HIF-1 and HIF-2 drive the metastatic potential of breast cancer cells to the lung in vivo (5-8). In addition, multiple mechanisms have been identified by which HIF promote breast to lung metastases including the enhanced secretion of proteins including lysyl oxidase (LOX) and LOX like proteins (LOXL2 and LOXL4) to prime the lung microenvironment for colonization (9, 10). HIF signaling also directly facilitates tumor extravasation to the lung through the upregulation of factors that promote tumor-endothelial cell adhesions or decrease endothelial cell-cell adhesion (7).

Response: Thank you for providing these additional references. We have further emphasized in the discussion that our findings are contradictory to a large body of work and include these studies on page 15.

One possible explanation for the observation that HIF-1 inactivation increases lung metastasis in this study may be that the HIF-1 knockout mice were sacrificed on average 20 days (in some cases >40 days) later than the control animals. This additional time may have allowed for additional time for the breast cancer cells to seed and/or grow in the lung. The authors noted that they normalized the metastatic tumor burden in the lung to endpoint primary tumor burden. However, normalization to time at sacrifice is necessary as metastatic potential is not always correlated with primary tumor size.

Response: This is an excellent point. We have now normalized the metastatic tumor burden in the lung to age at sacrifice in WT and Hif1 α ^{-/-} mice and included these data in Supplemental Figure 3D&F (included below). Normalizing the data to time at sacrifice did not change the outcome, and lung tumor burden remained significantly elevated in the Hif1 α ^{-/-} mice, indicating that this effect is independent of the additional time for the tumor cells to seed and/or grow in the lung. We have updated this in the results on page 8 and discussion on page 16.

References:

1. Woelfle U, Cloos J, Sauter G, Riethdorf L, Janicke F, van Diest P, Brakenhoff R, Pantel K. Molecular signature associated with bone marrow micrometastasis in human breast cancer. *Cancer research*. 2003;63(18):5679-84. Epub 2003/10/03. PubMed PMID: 14522883.
2. Cox TR, Rumney RM, Schoof EM, Perryman L, Hoye AM, Agrawal A, Bird D, Latif NA, Forrest H, Evans HR, Huggins ID, Lang G, Linding R, Gartland A, Erler JT. The hypoxic cancer secretome induces pre-metastatic bone lesions through lysyl oxidase. *Nature*. 2015;522(7554):106-10. Epub 2015/05/29. doi: 10.1038/nature14492. PubMed PMID: 26017313; PMCID: 4961239.
3. Hiraga T, Kizaka-Kondoh S, Hirota K, Hiraoka M, Yoneda T. Hypoxia and hypoxia-inducible factor-1 expression enhance osteolytic bone metastases of breast cancer. *Cancer research*. 2007;67(9):4157-63. Epub 2007/05/08. doi: 10.1158/0008-5472.CAN-06-2355. PubMed PMID: 17483326.
4. Dunn LK, Mohammad KS, Fournier PG, McKenna CR, Davis HW, Niewolna M, Peng XH, Chirgwin JM, Guise TA. Hypoxia and TGF-beta drive breast cancer bone metastases through parallel signaling pathways in tumor cells and the bone microenvironment. *PloS one*. 2009;4(9):e6896. Epub 2009/09/04. doi: 10.1371/journal.pone.0006896. PubMed PMID: 19727403; PMCID: 2731927.
5. Lu X, Yan CH, Yuan M, Wei Y, Hu G, Kang Y. In vivo dynamics and distinct functions of hypoxia in primary tumor growth and organotropic metastasis of breast cancer. *Cancer research*. 2010;70(10):3905-14. Epub 2010/05/06. doi: 10.1158/0008-5472.CAN-09-3739. PubMed PMID: 20442288; PMCID: 2872139.
6. Liao D, Corle C, Seagroves TN, Johnson RS. Hypoxia-inducible factor-1alpha is a key regulator of metastasis in a transgenic model of cancer initiation and progression. *Cancer research*. 2007;67(2):563-72. Epub 2007/01/20. doi: 67/2/563 [pii] 10.1158/0008-5472.CAN-06-2701. PubMed PMID: 17234764.
7. Zhang H, Wong CC, Wei H, Gilkes DM, Korangath P, Chaturvedi P, Schito L, Chen J, Krishnamachary B, Winnard PT, Jr., Raman V, Zhen L, Mitzner WA, Sukumar S, Semenza GL. HIF-1-dependent expression of angiopoietin-like 4 and L1CAM mediates vascular metastasis of hypoxic breast cancer cells to the lungs. *Oncogene*. 2012;31(14):1757-70. Epub 2011/08/24. doi: onc2011365 [pii] 10.1038/onc.2011.365. PubMed PMID: 21860410; PMCID: 3223555.
8. Goto Y, Zeng L, Yeom CJ, Zhu Y, Morinibu A, Shinomiya K, Kobayashi M, Hirota K, Itasaka S, Yoshimura M, Tanimoto K, Torii M, Sowa T, Menju T, Sonobe M, Kakeya H, Toi M, Date H, Hammond EM, Hiraoka M, Harada H. UCHL1 provides diagnostic and antimetastatic strategies due to its deubiquitinating effect on HIF-1alpha. *Nature communications*. 2015;6:6153. Epub 2015/01/24. doi: 10.1038/ncomms7153. PubMed PMID: 25615526; PMCID: 4317501.
9. Erler JT, Bennewith KL, Nicolau M, Dornhofer N, Kong C, Le QT, Chi JT, Jeffrey SS, Giaccia AJ. Lysyl oxidase is essential for hypoxia-induced metastasis. *Nature*. 2006;440(7088):1222-6. Epub 2006/04/28. doi: 10.1038/nature04695. PubMed PMID: 16642001.
10. Wong CC, Gilkes DM, Zhang H, Chen J, Wei H, Chaturvedi P, Fraley SI, Wong CM, Khoo

US, Ng IO, Wirtz D, Semenza GL. Hypoxia-inducible factor 1 is a master regulator of breast cancer metastatic niche formation. *Proceedings of the National Academy of Sciences of the United States of America*. 2011;108(39):16369-74. Epub 2011/09/14. doi: 10.1073/pnas.1113483108. PubMed PMID: 21911388; PMCID: 3182724.

Reviewer #2 (Remarks to the Author):

In the paper "Hypoxia inducible factor signaling in breast tumors controls spontaneous tumor dissemination in a site-specific manner", Todd and colleagues used a PyMT-driven spontaneous mouse mammary carcinoma model with mammary specific Hif1a, Hif2a, or Vhl knockout to investigate whether HIF signaling controls site-specific breast tumor metastasis. Although this work provides some interesting observations, the data presented is not solid enough to back up the conclusion.

1. It is will established that Vhl degradation results in Hif1 and Hif2 protein stabilization upon hypoxia. It is surprising to see no examination of Hif1a and Hif2a protein level in the primary tumors, bone and lung metastasis using Western blot or IHC. This data is particular important when performing experiments using the Vhl knockout mice.

Response: Thank you for your comments and we understand the concern. We have attempted to stain for Hif1 α and Hif2 α protein levels in the tumor samples using multiple antibodies but were unable to detect signal above background. Alternatively, to further confirm that Hif signaling is up-regulated in the Vhl knockout mice, we have now performed qPCR on *Vegfa* (Hif1/2 α target), *Pgk1* (Hif1/2 α target), *Glut1* (Hif1 α target), *Pdgfb* (Hif1 α target), and *Tek* (Hif2 α target) and confirmed that these are all elevated in the primary tumors (Figure 6G-K and included below). While these data do not definitively show stabilization of the Hif proteins, they do indicate sustained biological activity of the proteins *in vivo*. We have added this to the results on page 12. We have also confirmed that Vhl was recombined *in vivo* (Figure 6F). While we were analyzing Vhl^{f/f} and Vhl^{-/-} tumor samples for expression of additional HIF target genes, we were able to re-analyze one sample that originally had poor quality cDNA and was excluded from *Vhl*, *Vegfa*, *Cxcr4*, and *Pthlh* analysis. The addition of this sample to complete that dataset shifted the p-value for *Cxcr4* expression from p=0.04 to p=0.077. This does not alter our conclusion, but we have amended the wording in the results section on page 13 to reflect this change.

2. Follow up from the previous point, *Vhl* not only regulates Hifs but also numerous factors in the cancer cells (as the authors described in the discussion). Since the authors used *Vhl* knockout as a Hif1/Hif2 gain-of-function model, it is necessary to use *Vhl*/Hif1a or *Vhl*/Hif2a double-knockout mice to demonstrate that the effects observed in *Vhl* knockout is Hif1 or Hif2 specific.

Response: We agree that this experiment would be ideal to assess whether the mechanism downstream of *Vhl* is Hif1 α or Hif2 α specific. We unfortunately do not have funding to support creating these additional mouse strains. We have added a comment to the discussion on page 19-20 to indicate that these experiments are necessary to confirm whether the mechanism is mediated by Hif1 α or Hif2 α .

3. Most of the results in this manuscript showed no differences between the control and knockout. It could be due to the sample collection time. Mice were sacrificed when tumor size reached 1 cm in diameter in the mammary gland. Maybe a longer time (when the tumor reach 2 cm) will show a more dramatic effect? Or a shorter time point can capture the difference between the knockout and the control? The authors are encouraged to perform a time course experiment to detail characterize the kinetic difference in bone and lung metastasis in the knockout animal compared to the control.

Response: The sample collection times for these studies are similar to the sample collection times previously published for similar models (Liao, et al. Cancer Res. 2007). We agree it is an excellent idea to examine tumors at a later time point; however, our IACUC does not

allow tumors to exceed 1cm, and since the tumors are multi-focal we cannot ethically perform tumor removal/survival surgeries. We were also required to cull down and cryopreserve all three mouse strains last year due to our institution's COVID-19 requirement that we reduce animal colonies. While we would like to perform the shorter time course studies, it will be difficult to identify significant differences between control and knockout groups due to the low levels of disseminated tumor burden that are reported in the early progression of the PyMT model. This would require re-deriving the colonies, and to ensure robust statistical analyses for all strains, the combined collection of several hundred mice. The breeding scheme for these mice is also complicated, given that the females cannot carry the PyMT transgene, and only female progeny are collected. Unfortunately, we do not have the resources required to perform these studies.

4. In figure 4, expression of EMT-related genes in primary tumors were examined and no difference was found between the control and Hif1a knockout group. This is not surprising as HIF2 and other hypoxia induced signaling pathways are known to upregulate EMT. What's confusing is why the authors used hypoxia gene signature identified in kidney cancer (Li et al) to correlate with bone and lung metastasis in breast cancer? Please check this reference for 42-gene hypoxia signature in breast cancer (PMID: 30037853).

Response: Thank you for bringing this more recent study to our attention. We have removed the correlation to the kidney cancer study and replaced those panels with the breast cancer hypoxia gene signature correlation in Figure 4D&E (included below) and updated the results on page 9. The updated hypoxia gene signature for breast cancer still positively and significantly correlates with the bone metastasis and lung metastasis signatures.

5. On page 9, line 168: "the increased lung tumor burden is due to lung microenvironment-specific factors promoting outgrowth of disseminated tumor cells." If the Hif knockout is mammary tissue specific, what will be the lung specific factors between the control and knockout mice that promote the metastatic tumor growth?

Response: Apologies for the confusion. We have clarified this statement to say "...the increase in lung tumor burden may be due to altered Hif1 α -/- tumor cell response to signals from the lung microenvironment," on page 9. We do not believe that the lung specific factors are changing, just the tumor cell response to those factors.

Minor:

1. In the discussion section, there is a typo on line 259: It should be "Hif1a" not "Hi12a".

Response: Thank you, this has been corrected on page 14.

2. The papers cited in the introduction are not up to date. There are many recent research and review discussing Hifs and cancer metastasis. The authors should cite more recent papers.

Response: We have added more recent references to the introduction (References 1, 6, 21, 29, 31, and 39).

Reviewers' comments:

Reviewer # 1 (Remarks to the Author):

-The authors have included PTHrP mRNA expression data in Figure 8 to provide data to implicate altered PTHrP in the decreased trabecular bone. However, there were no significant differences observed in PTHrP expression between any of the groups. Additionally, in Figure 8, Cxcr4 expression is examined and there are no significant differences shown. Therefore, the model in Fig 8G needs to be revised to reflect the conclusions that are significant in the manuscript. In addition, the title for Figure 8 needs to be revised as well as the abstract "In contrast, known metastasis regulators such as CXCR4 and parathyroid hormone related protein (PTHrP) appear to regulate bone dissemination in response to VHL or HIF2 α expression in the primary tumor" as these conclusions are not supported by the data.

-The authors have addressed other concerns.

Reviewer #2 (Remarks to the Author):

The authors have addressed my comments and revised the manuscript accordingly. I have no further comments.

Response to Reviewers

Reviewer # 1 (Remarks to the Author):

-The authors have included PTHrP mRNA expression data in Figure 8 to provide data to implicate altered PTHrP in the decreased trabecular bone. However, there were no significant differences observed in PTHrP expression between any of the groups. Additionally, in Figure 8, Cxcr4 expression is examined and there are no significant differences shown. Therefore, the model in Fig 8G needs to be revised to reflect the conclusions that are significant in the manuscript. In addition, the title for Figure 8 needs to be revised as well as the abstract "In contrast, known metastasis regulators such as CXCR4 and parathyroid hormone related protein (PTHrP) appear to regulate bone dissemination in response to VHL or HIF2 α expression in the primary tumor" as these conclusions are not supported by the data.

Response: Thank you for reviewing our revised manuscript and constructive feedback. We have now altered the model in Fig 8G by removing PTHrP and CXCR4 and updated the results accordingly (page 13), revised the title for Figure 8 (page 54), and removed the statement referenced above from the abstract. We have also made modifications to the discussion to better match the model proposed in Figure 8 (pages 18-19).

-The authors have addressed other concerns.

Response: Thank you, we appreciate the feedback.

Reviewer #2 (Remarks to the Author):

The authors have addressed my comments and revised the manuscript accordingly. I have no further comments.

Response: Thank you, we appreciate the feedback.